health and disease and epidemiology/e-science

national accounts, transaction data, consumption, COVID-19

**Author for correspondence:**
Stephen Hansen
e-mail: stephen.hansen@imperial.ac.uk

# Tracking the COVID-19 crisis with high-resolution transaction data

Vasco M. Carvalho[1,2], Juan R. Garcia[3], Stephen Hansen[4], Álvaro Ortiz[3], Tomasa Rodrigo[3], José V. Rodríguez Mora[2,5] and Pep Ruiz[3]

[1]Faculty of Economics, University of Cambridge, Cambridge, UK
[2]Alan Turing Institute, UK
[3]BBVA Research, Banco Bilbao Vizcaya Argentaria SA, Madrid, Spain
[4]Imperial College Business School, London, UK
[5]University of Edinburgh, Edinburgh, UK

VMC, 0000-0002-6128-9157; SH, 0000-0002-8376-6292

Payments systems generate vast amounts of naturally occurring transaction data rarely used for constructing official statistics. We consider billions of transactions from card data from a large bank, Banco Bilbao Vizcaya Argentaria, as an alternative source of information for measuring consumption. We show, via validation against official consumption measures, that transaction data complements national accounts and consumption surveys. We then analyse the impact of COVID-19 in Spain, and document: (i) strong consumption responses to business closures, but smaller effects for capacity restrictions; (ii) a steeper decline in spending in rich neighbourhoods; (iii) higher mobility for residents of lower-income neighbourhoods, correlating with increased disease incidence.

## 1. Introduction

Every day, banks, payments systems providers and other financial intermediaries record and store massive amounts of individual transaction records arising from the mundane course of economic life. As more and more of the world's trade and exchange activity is intermediated on platforms underpinned by digital technology, real-time, high-resolution transaction data is likely to continue to grow rapidly.

While there is broad agreement among national statistical agencies that unstructured transaction data will play an increasingly prominent role in twenty-first century national accounting (see Bean [1], Abraham *et al.* [2] and Jarmin [3]), national statistical agencies, academics and policy-makers still largely rely on more traditional structured survey data and

slow-moving national accounts updates.[1] Partly, this reluctance reflects concerns regarding the accuracy and representativeness of transaction data. Indeed, traditional economic measurement relies heavily on centrally administered, carefully designed surveys conducted with representative subsamples of the population. By contrast, transaction data arises through the decentralized activity of millions of economic agents. How then do such data compare to national accounts? Which potential biases and distortions exist in indices built from transactions, and what additional insights can they bring? While there is a reasonable expectation that economists and government agencies will have increased access to large-scale transaction datasets in the near future, extensive validation against available official statistics is needed in order for transaction data to fulfil its promising role in national accounting.

The first contribution of this paper is to analyse these issues in the context of the universe of credit and debit card transactions mediated by a large global bank, Banco Bilbao Vizcaya Argentaria S.A. (BBVA). Our data consist of the universe of transactions collected from BBVA cardholders and BBVA-operated point-of-sale in Spain, accounting for 2.1 billion transactions.[2] We explore the properties of the data along three different dimensions: as a high-frequency coincident indicator for aggregate and subnational consumption; as a detailed household consumption survey; and as a mobility index.

In each case, we show that card spending captures some but not all of the relevant information in the analogous official data series, but nevertheless acts as an informative proxy along comparable cuts of official data. This then allows one to make further cuts into the spending data to obtain insight unavailable using external series alone.

Our second contribution is to show how this transaction data, once validated, offers several policy-relevant lessons from the first Spanish lockdown—one of the world's harshest—that are relevant for the numerous countries currently re-entering lockdown. We use the data along each of the three dimensions above to obtain valuable, but otherwise largely hidden, lessons related to the effects of the pandemic and lockdown policies.

First, we exploit subnational high-frequency expenditure data in tandem with systematic changes in lockdown policies across spatial units to evaluate the differential effects of those policies. We show that restrictions of activity that work through limiting capacity and customer density have only a mild effect on expenditure, in particular, when compared with the effect of forcing the closure of large retail establishments.

Second, we exploit the transaction data as a detailed consumption survey, which allows us to track changes in the composition of consumption and the structure of consumption across income classes. We document that residents of the richest postal codes suffered the largest declines in expenditure during lockdown. Furthermore, we show that this is explained because lockdown restrictions, by their very nature, affect more predominantly the pattern of conspicuous consumption prevalent in wealthier individuals.

Third, we show that expenditure in transportation correlates exceedingly well with external mobility measures, and that during the lockdown the mobility of the rich was substantially smaller than that of the poor. Moreover, we also show that differential mobility patterns predict heterogeneity in the incidence of the pandemic across income groups.

The main methodological contribution our paper makes is to benchmark card spending data against external series to assess its plausibility to conduct analysis of granular economic activity. Datasets arising from card spending and point-of-sales terminals are currently and will probably remain one of the most commonly available transaction datasets. The comparison exercises we conduct, and the strengths and weaknesses of the data we identify, are hence more broadly relevant beyond BBVA.

The main applied contribution of the paper is to document expenditure adjustments during the COVID-19 pandemic. Relative to this large and fast-expanding literature, we encounter some common patterns. Thus, like [7,8] in US studies, and [9] for the UK, we find that higher-income groups witnessed the largest fall in expenditures during the crisis. Our analysis of cross-category expenditure reallocation during the crisis echoes findings elsewhere in the literature, for example in [10] for France; [11] for Portugal; [12] for the UK; and [13,14] for the USA. Furthermore, our analysis of the effects of

---

[1]Important exceptions include [4–6], which use data from financial apps to test consumption smoothing theories.

[2]Since BBVA is a global bank, it generates several billion more transactions across other countries in which it has a large market share, for example Turkey, Mexico and the Southern USA. An earlier version of this manuscript included discussion of the global time series, which can be downloaded here https://www.bbvaresearch.com/en/special-section/charts/ but which we omit for space constraints.

lockdown and its easing complements that in [15]. The latter argues for the importance of behavioural adjustments in expenditure patterns, responding to local disease dynamics even in the absence of lockdown policies. Consistent with this, we find local disease incidence to be a driver of expenditure growth changes, even when controlling for different levels of lockdown restrictions across space. Unlike [15], we are able to additionally document the significant effects of different lockdown restrictions, even when controlling for local disease incidence. Finally, like [16,17], we explore the relation between mobility and disease incidence. Relative to that contribution, we show that in the absence of direct mobility proxies, card transactions in transportation categories can be used as a mobility proxy at narrow geographical and socio-economic status levels of analysis.

# 2. Results

We organize the results by first validating proxy measures derived from Spanish card data against external data in Spain, then applying the proxy to understand an important aspect of the COVID-19 crisis.

## 2.1. Transaction data as a high-frequency consumption proxy

### 2.1.1. Validation

We compare total spending via BBVA cards and point-of-sale (PoS) terminals with the national account household consumption series (Non-Durable Household Domestic Final Consumption) for every quarter since 2016. We also compare time series of spending at monthly frequency, and on specific components present in BBVA and national accounts. As detailed in the electronic supplementary material, we find that: (i) BBVA card expenditure series correlate highly both with aggregate national-accounts consumption (correlation of 0.874), and within narrowly defined consumption categories series where official data is available, in particular expenditures at gas stations (correlation of 0.784); (ii) that nevertheless, both at the aggregate and sector level the BBVA series is more volatile than the official series. The likely cause of the latter is that card data does not cover stable household expenses—such as rents, school fees, some utilities and subscription services—and that over long spans of time there are probably extensive margin movements, reflecting entry and exit of clients, cards and PoS in the BBVA sample.

We next validate spending data in the cross-section of geographical units. There are no official subnational consumption measures in Spain, so instead we compare BBVA spending to official data on income in Spanish provinces (52 in total) and Madrid postal codes. The correlations are extremely high: 0.975 across provinces and 0.923 across postal codes (further details in electronic supplementary material).

The conclusion is that card spending captures important patterns across space and time in national accounts data, albeit with more noise in the latter than the former.

### 2.1.2. Effects of lockdown and its easing

With many major European economies again facing extended lockdowns due to a resurgence in COVID-19 cases, the optimal balance between economic activity and public health is again of paramount importance. We next use the imposition of the first lockdown in Spain in March 2020 and its subsequent, progressive easing to draw lessons for managing restrictions going forward.

The electronic supplementary material contains background information on the development of COVID-19 and Spanish government policy responses during March–June 2020, and here we provide a brief summary. A national lockdown was first imposed beginning on 15 March in response to rapidly growing infections. The measures were among the harshest in the world and resulted in the suspension of all but essential economic activity. After a subsequent fall in cases, the government began *Phase 0* easing on 4 May, which permitted small retail stores to operate under strict social distancing guidelines. This first easing stage applied uniformly to all regions in Spain, but further easing was staggered across provinces.

On 11 May, some provinces entered *Phase 1* which allowed for larger retail spaces (but not superstores and malls) to reopen at restricted capacity and for outdoor commercial activity (including restaurants) to resume. *Phase 2* then began on 25 May for some provinces and lifted all size restrictions on commercial

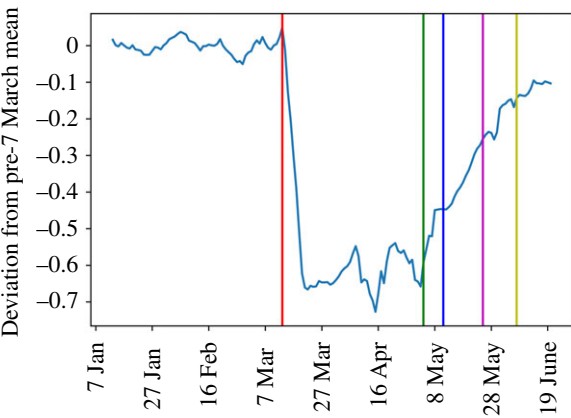

**Figure 1.** Moving average (7 day, uncentred) of Y-o-Y growth of expenditure from BBVA series for Spain (aggregate). The vertical lines indicate the timing of events. The first (red) vertical line is drawn on 13 March, the day prior to the announcement of lockdown. The second one is 4 May (start of Phase 0), when easing started nationwide. The third vertical line stands for 11 May (start of Phase 1), when provinces started to differentiate in the intensity of the lockdown, some of them easing lockdown faster than others. The remaining lines are drawn on 25 May (start of Phase 2 for some provinces) and 8 June (start of Phase 3). The series is normalized by the Y-o-Y growth before 7 March.

activity (including malls) and some indoor commercial activities, while still keeping capacity caps in place. *Phase 3* began on 8 June and relaxed further these capacity limits.[3]

Figure 1 plots aggregate expenditure growth in Spain over this period, normalized by average year-on-year (Y-o-Y) growth prior to 8 March. Expenditure growth fell abruptly on the day of lockdown, by about 60 percentage points (p.p.) and remained depressed at that level until early May, when easing of lockdown ensued. The aggregate data is also suggestive of a recovery starting with the nationwide enactment of *Phase 0*. By 21 June, when our data end, expenditure growth in Spain is only a few percentage points off its pre-COVID-19 average, denoting a near complete recovery in expenditure.

The staggered adoption of easing phases across provinces, combined with spending data at the day and province level, provides a unique opportunity to study consumption reactions to different kinds of economic restrictions. Figure 2a plots the average Y-o-Y expenditure growth for the provinces which eased into Phase 1 on 11 May (in orange) against the average growth for those provinces that remained in the more restrictive Phase 0 (in blue). Figure 2b,c plots the corresponding event-study graphs centred around 25 May and 8 June, when some provinces further eased into Phase 2 and Phase 3, respectively. The easings into Phases 1 and 2 appear to be on average associated with higher spending for switchers versus stayers. On the other hand, the easing into Phase 3 has a much less marked impact. This provides evidence that shop openings generate more economic impact than the lowering of capacity restrictions. To the extent that capacity restrictions provide public health benefits, this provides a strong rationale for maintaining them in place whenever even moderate infection risk is present.

While this initial analysis provides suggestive evidence, the fact remains that different Spanish provinces are: (i) selected into treatment based, at least partly, on disease incidence and (ii) differ along a host of observable and unobservable characteristics. To at least partly address this issue, we now turn to regression analysis. In table 1, we present panel regressions of the daily provincial Y-o-Y growth of expenditure on lockdown-phase and easing dummies i.e. binary variables for each province and period, which take a value of one if that particular province is classified in a particular phase of the lockdown—or lockdown easing—a given calendar day and zero otherwise. Note further that, as discussed above, for the week immediately preceding the lockdown, the lockdown itself and Phase 0 of lockdown easing, all provinces move in lockstep, so these categorical variables display the same time pattern for all provinces. Instead, for Phases 1, 2 and 3, the time pattern is province-specific, depending on when a particular province advanced to the later lockdown easing phases. Throughout standard errors are clustered at the province level.

---

[3]A further *Phase 4* began on 21 June and represented a return to essentially normal economic activity, but we exclude this from our sample below due to too few days entering this period.

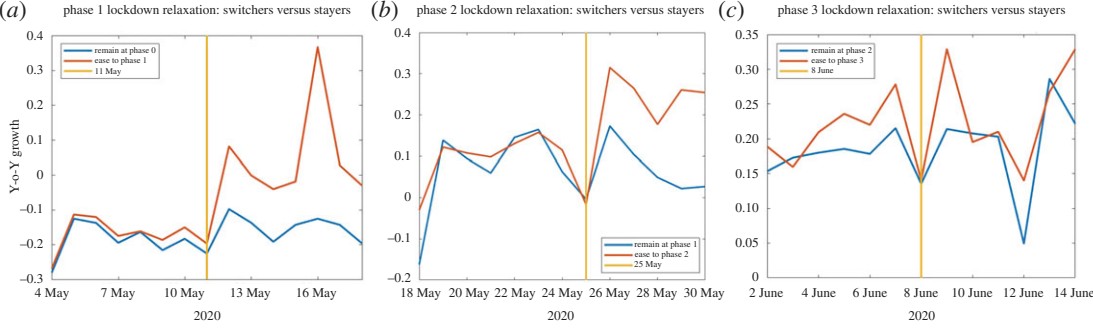

**Figure 2.** Event study graphs. (*a*) Average Y-o-Y expenditure growth for the provinces which eased into Phase 1 on 11 May (in orange) and average growth for provinces that stayed in the more restrictive Phase 0 (in blue). (*b*) As in panel (*a*), but centred around 25 May when some provinces eased into Phase 2 while others remained in Phases 0 and 1. (*c*) As in panel (*a*), but centred around 8 June, when some provinces eased into Phase 3 while others remained in previous Phases. All figures use deseasonalized data obtained as follows: we first regress our Y-o-Y province-level growth series on a full set of day of the week dummies. We then plot event-study graphs using de-seasonalized daily expenditure growth, centred around lockdown easing announcement days.

The first column gives the basic province-level time-series pattern in the data, as a function of the stage of lockdown and easing. In particular, we regress province Y-o-Y expenditure growth on a series of time dummy variables, where the omitted category is the period before 8 March one week before any official discussion of lockdown enactment. The reported coefficients can thus be read as the excess percentage point growth of average provincial expenditure, relative to pre-pandemic growth and as a function of the policy adopted at each stage of the pandemic.

It is clear that expenditures increased substantially (an average of more than 8 p.p. across provinces) in the week ahead of the lockdown, most likely in anticipation of it. The period of strict lockdown, with its associated restrictions on commercial activity, led to a large fall of about 60 p.p. in Y-o-Y growth of expenditure. These patterns are consistent with figure 1 where one observes that this expenditure contraction coincides with the beginning of lockdown and lasts as long as restrictions remain at their strictest level, up until 4 May.

Likewise, it is apparent that the initial easing of the restrictions—Phase 0, applied nationally—coincides with a sudden increase of activity. While different provinces remained at this institutional stage (and level of restrictions) for different lengths of time, the average value of Y-o-Y growth of expenditure is on average about 12 p.p. higher than in the preceding, strict lockdown, period.

The point estimates in column (1) indicate that further easing of restrictions is associated with further substantial improvements of expenditure growth, Y-o-Y growth being 'only' 8 p.p. lower than its pre-lockdown value by the time a province reaches Phase 3. Overall, based on these simple means, the Phases 1 and 2 easings which opened progressively larger retail spaces and hospitality (albeit still under capacity restrictions) seem to contribute the most to a strong expenditure recovery.

In columns (2), (3) and (4) of table 1, we additionally control, respectively, for differential disease dynamics across provinces, province fixed effects and both together. Daily provincial incidence of COVID-19 (measured as the number of new cases per 1000 habitants), provides a first attempt at dealing with the basic endogeneity issue: the policy decision to ease restrictions depends on the incidence of COVID-19 at the province level, and provinces with less incidence should be expected to perform better, even in the complete absence of restrictions to activity. Consumption expenditure indeed seems affected by the incidence of the disease, even conditional on the de jure restrictions in place. Province fixed effects additionally control for systematic differences across provinces, such as in income, population density, rural/urban prevalence, which can be assumed to be fixed (or at least slowly varying) at the daily frequency. Across these specifications, the point estimates on the effects of lockdown and subsequent easing phases are essentially unchanged.

Finally, in column (5) of table 1, we present difference-in-differences estimates with province and day fixed effects and stage-of-easing-specific dummy variables. Column (6) additionally controls for the daily incidence of the pandemic at the province level. Note that, due to the inclusion of time fixed effects our estimates are now identified out of differences in the timing of (the easing of) restrictions at the province level, thus yielding a standard difference-in-differences set-up with (i) variation in treatment timing across units and (ii) multiple treatments. Note also that, relative to

**Table 1.** Panel regressions of daily provincial Y-o-Y growth of expenditure on phase of the lockdown and easing-date province specific dummies. Column (2) controls for daily disease incidence at the province level. Columns (3) and (5) add provincial fixed effects and provincial and day fixed effects, respectively. Columns (4) and (6) add disease incidence controls. Standard errors are clustered at the province level. BBVA data to 21 June. Daily incidence of COVID-19 in each province obtained from the Spanish Health Ministry https://cnecovid.isciii.es/covid19/#documentaci%C3%B3n-y-datos.

| | daily Y-o-Y expenditure growth by province | | | | | |
| --- | --- | --- | --- | --- | --- | --- |
| | (1) | (2) | (3) | (4) | (5) | (6) |
| week before lockdown | 0.0844*** | 0.111*** | 0.0844*** | 0.102*** | | |
| | (0.00837) | (0.0124) | (0.00839) | (0.00947) | | |
| lockdown | −0.598*** | −0.570*** | −0.598*** | −0.580*** | | |
| | (0.0143) | (0.0190) | (0.0143) | (0.0154) | | |
| lockdown easing | | | | | | |
| Phase 0 | −0.478*** | −0.475*** | −0.471*** | −0.471*** | | |
| | (0.0186) | (0.0183) | (0.0174) | (0.0174) | | |
| Phase 1 | −0.263*** | −0.262*** | −0.264*** | −0.262*** | 0.108*** | 0.109*** |
| | (0.0164) | (0.0166) | (0.0159) | (0.0160) | (0.0181) | (0.0186) |
| Phase 2 | −0.125*** | −0.125*** | −0.127*** | −0.127*** | 0.210*** | 0.211*** |
| | (0.0148) | (0.0148) | (0.0143) | (0.0143) | (0.0285) | (0.0295) |
| Phase 3 | −0.0756*** | −0.0763*** | −0.0815*** | −0.0801*** | 0.242*** | 0.245*** |
| | (0.0248) | (0.0246) | (0.0207) | (0.0207) | (0.0394) | (0.0408) |
| daily COVID incidence | | −0.2802** | | −0.183*** | | −0.0411 |
| | | (0.1153) | | (0.0494) | | (0.0490) |
| province fixed effects | N | N | Y | Y | Y | Y |
| day fixed effects | N | N | N | N | Y | Y |
| N | 8378 | 8378 | 8378 | 8378 | 8378 | 8378 |
| $R^2$ | 0.431 | 0.434 | 0.526 | 0.527 | 0.753 | 0.753 |

***$p < 0.01$; **$p < 0.05$.

the previous specifications, the omitted category is now 'Phase 0', the last common policy baseline across all provinces and therefore the interpretation of the coefficients changes. For example, estimates pertaining to Phase 1 now give the percentage *growth* in expenditures for provinces that proceeded to this lockdown easing stage—at whatever calendar date they may have done so— relative to remaining at Phase 0 for longer (for further discussion on interpretation and references on this estimator, see our Methods §4).

The estimates we obtain are nevertheless similar to the ones obtained previously. Thus, we again observe that Phases 1 and 2 induce sizeable recoveries in expenditure growth by enlarging the set of establishments available to consumers. At the same time, the intensive margin easing of capacity restrictions associated with Phase 3 does not generate a statistically significant differential effect. Furthermore, these conclusions are unaffected by the inclusion of province-level disease dynamics and, as we show in the Methods section below, are also robust to further checks related to the possible endogeneity of the timing of lockdown easing.

Finally, these robust correlations notwithstanding, we end this section with a word of caution when interpreting these estimates as the true 'causal effect' of lockdown policies. This is because, as is well known, identification of causal effects in our context would require province-level lockdown policies and their timings to be 'as good as random', at least conditional on time and province fixed effects and, possibly, other relevant time-varying province-level covariates. In particular, while our most demanding specification above attempts to account for all of these, we cannot rule out the presence of other unobserved, time-varying, province-level conditions which (i) may have influenced selection into treatment—beyond the province-specific prevalence of COVID; for example evolving socio-economic considerations by the Spanish government—and/or (ii) have had a bearing on expenditure

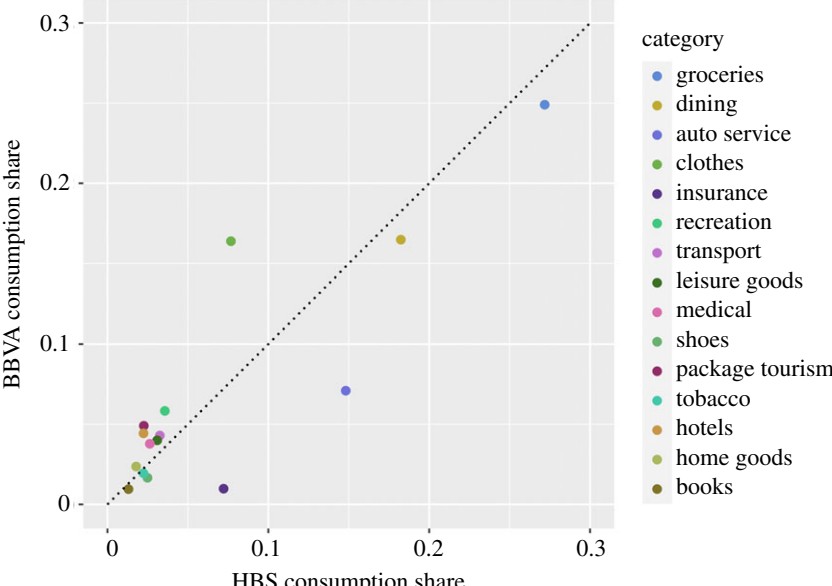

**Figure 3.** Consumption share comparison in matched ECOICOP product categories.

decisions of households such as the evolution of province-specific expectations of disease prevalence which, in turn, may lead to behavioural expenditure responses that go beyond the particular de jure lockdown regime and may not be accounted for by province-level disease prevalence. Thus, to the extent that these time-varying province-level unobservables were operational during lockdown easing in Spain, our estimates may be biased—in either direction—relative to the true causal effect.

## 2.2. Transaction data as a granular consumption survey

### 2.2.1. Validation

National statistics organizations traditionally measure household consumption baskets with representative spending surveys. On the other hand, transaction data derived from card transactions typically contain associated metadata which allows a breakdown of expenditure across goods and services categories. Can these two sources of data be bridged? Can metadata on card transactions stand in for nationally representative consumption surveys?

In the electronic supplementary material, we compare in detail household spending across categories as measured by the official Spanish Household Budget Survey (HBS) and by the BBVA dataset, which breaks purchases into one of 77 distinct categories. The two data sources have distinct categorizations, which require a manual match; in total, we find matching categories for 65% of BBVA spending. Figure 3 plots consumption shares in the matched categories in both datasets, which have a correlation coefficient of 0.865.

In a second validation exercise, we consider the subsample of BBVA transactions that involve a BBVA debit or credit card, in which case we have information on the consumer's demographic characteristics. As we detail in the electronic supplementary material, the share of consumption per age and education groups aligns remarkably well between BBVA data and HBS.

Finally, we tabulate total BBVA debit and credit card spending by Madrid postal code, and use postal code income as a proxy for household income. The allocation of consumption across categories according to income derived from BBVA data also aligns exceedingly well with the one observed in HBS.

These three validation exercises demonstrate that information derived from BBVA purchase categories aligns relatively well with information from the HBS along comparable cuts of data, a fact we can use to document the allocation of spending in real time during the onset of the COVID-19 crisis.

### 2.2.2. Composition of consumption in the lockdown

Our first application of using card spending as a consumption survey is to study the spending reallocation induced by the Spanish lockdown (15 March to 4 May). The electronic supplementary

**Table 2.** Best and worst performing categories of expenditure by market share post-lockdown growth. In bold, categories restricted during the lockdown. (ITV: Inspecciòn Tècnica de Vehiculos.)

| top 10 sectors in market share growth | | bottom 10 sectors in market share growth | |
| --- | --- | --- | --- |
| (decreasing order of gain) | growth | (decreasing order of loss) | growth |
| food: small stores | 2.24853 | **fashion** | −0.97797 |
| tobacco store | 2.22432 | **pubs and disco clubs** | −0.93504 |
| mobile phone credit | 2.06751 | **furniture and decoration chains** | −0.932594 |
| supermarkets | 1.98371 | **leather shops** | −0.93121 |
| hypermarkets | 1.67307 | **shoe shops** | −0.928647 |
| pharmacy and parapharmacy | 1.52951 | **toys: chains** | −0.920665 |
| gifts and donations | 1.12815 | **massage and personal care** | −0.894873 |
| insurance | 0.835929 | **fashion: small shops** | −0.892908 |
| veterinary and pets | 0.719036 | **restaurants** | −0.883958 |
| newspapers and press | 0.668963 | **automobile inspection (ITV)** | −0.871738 |

material lists the 77 BBVA spending categories, and identifies the categories that were directly subject to lockdown restrictions which include a broad set of non-essential shopping categories as defined by the Spanish government.

Table 2 lists the top 10 and bottom 10 spending categories according to the evolution in market share before and after lockdown; categories directly affected by lockdown measures are in bold, which (perhaps unsurprisingly) constitute all of the bottom categories and none of the top categories. More notable are the enormous shifts in spending in this period, with some categories collapsing nearly entirely while others increase by 100% or more their market share. The goods and services with market share growth in lockdown relate to basic necessities (such as food), or have very low demand elasticity (such as tobacco). All of them were deemed critical sectors, and remained open for business during the lockdown, albeit with restrictions on capacity and customer density at any given point in time.

Figure 4 provides visual evidence of these spending shifts by plotting the market share across 18 broad spending categories that combine the 77 disaggregated categories. These shares are quite stable up until the week preceding the national lockdown, when a clear reallocation pattern emerges: spending on food and in 'hypermarkets' (i.e. large superstores) grows considerably, and these two categories alone make up over half of all expenditure by late March. At the same time, other sectors (such as fashion and leisure and entertainment) collapse entirely. Moreover, in the same manner that aggregate spending recovered quickly once the easing of restrictions began, the composition of consumption returns steadily to pre-lockdown allocations following the entry in the 'phase 0' of the easing period, on 4 May. We provide further time-series figures in the electronic supplementary material to study the evolution of the disaggregated categories.

### 2.2.3. Dynamics of aggregate consumption across income groups during the lockdown

The shift in consumption during lockdown masks important underlying heterogeneity with respect to income, which our card data allow us to explore in detail using expenditure patterns by different Madrid postal codes. Here, we measure spending of BBVA cardholders who have a registered address within a given postal code, and exclude PoS spending since we do not observe the home address of non-BBVA cardholders.

In table 3, we present the categories that during 2019 were most positively and negatively correlated with postal-code income *per capita*. One observes a pattern whereby higher-income groups consume goods associated with leisure and market production, while lower-income groups purchase more necessities and engage in home production. Marked in bold are those categories whose consumption was restricted during lockdown. Goods associated with the higher-income groups are relatively more affected by lockdown restrictions, which suggests that the consumption basket of higher-income groups became more like that of the poor during lockdown.

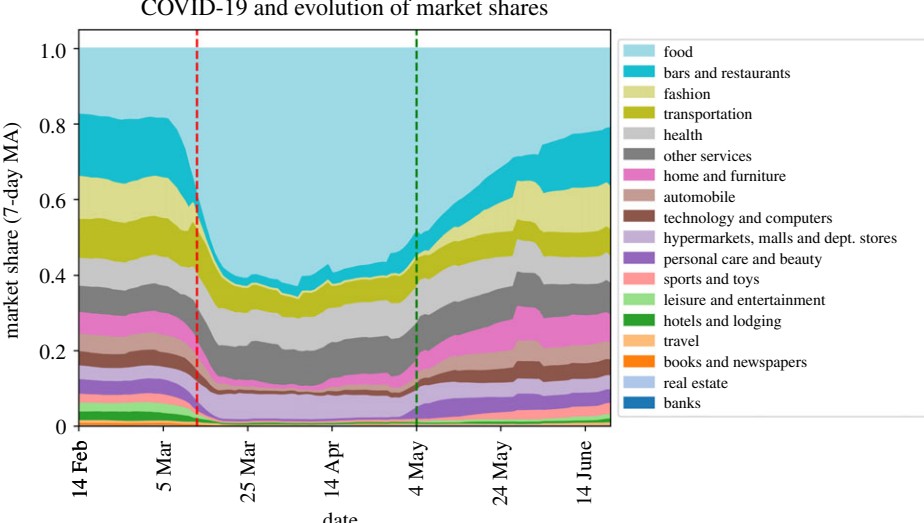

**Figure 4.** Evolution of the share of offline spending across categories. Red dashed line = national lockdown begins; green dashed line = national lockdown begins to ease. Shares are computed as seven-day moving averages (MA).

**Table 3.** Categories more positively and negatively correlated with average income across Madrid postal codes. In bold, categories restricted during the lockdown.

| high-income categories | | low-income categories | |
|---|---|---|---|
| category | corr. with income | category | corr. with income |
| taxi | 0.67 | gas stations | −0.48 |
| **sports** | 0.62 | supermarkets | −0.35 |
| **beauty and hairdressers** | 0.58 | **car technical inspection** | −0.35 |
| **restaurants** | 0.58 | telephony | −0.26 |
| **parking** | 0.53 | **DIY: small retail** | −0.25 |
| **fashion: small retail** | 0.42 | insurance | −0.25 |
| **mid- and long-distance trains** | 0.41 | tobacco | −0.23 |
| pharmacy | 0.40 | **auto sales/repair/parts** | −0.23 |
| travel agency: physical location | 0.38 | veterinary | −0.22 |
| **bars and coffee shops** | 0.37 | miscellaneous | −0.18 |

The implications of the alternative consumption baskets consumed by different income groups can be seen in figure 5, which plots a moving average of expenditure growth for Madrid postal codes binned by quintile according to income *per capita*. The sharpest declines in spending during lockdown concentrate in the richest postal codes, which is consistent with the rich being unable to consume their normal goods basket due to restrictions.

In the electronic supplementary material, we perform more formal statistical analysis in order to quantify these effects more rigorously, and control for disease dynamics that might also drive neighbourhood-level spending. These regressions not only confirm that wealthier neighbourhoods were the ones experiencing the largest fall in expenditure. They additionally suggest that areas more affected by the pandemic had larger declines in expenditure.

## 2.3. Transaction data as a real-time mobility proxy

### 2.3.1. Validation

The final aspect of information that we focus on from card spending is mobility patterns. Mobility and its determinants have become major issues during the COVID-19 pandemic due to the control of movement

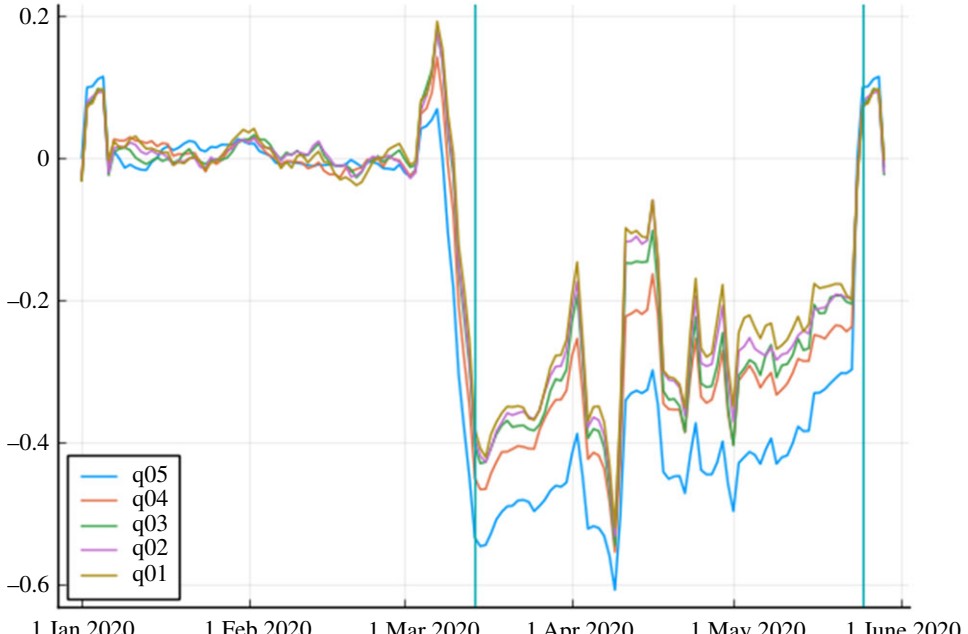

**Figure 5.** Y-o-Y growth rate of expenditure in Madrid's postal codes during 2020 by postal code average income (in quintiles). Normalized by the average Y-o-Y growth before 8 March 2020. The two vertical lines indicate (i) the lockdown day (15 March) and (ii) the day the whole of Madrid went into Phase 1 of the easing process (25 May).

being a key goal of social distancing policies (e.g. [18,19]), but mobility studies typically rely on data captured from users' mobile phones. In countries like the USA, these data are available at fairly disaggregated spatial units and also contain information on user characteristics. In other countries, such data are much rarer and so alternative mobility proxies are important to find.

Besides shopping for essential goods, the main source of mobility during Spain's lockdown was commuting for work. We use card data to measure this by considering BBVA spending categories that relate directly to transportation: 'bus trips'; 'gas stations'; 'parking'; 'tolls'; 'taxi'; 'urban transport'; and 'trains'. To validate this as a travel-to-work measure, we compare transportation spending growth against the 'work places' and 'transit' stations categories from Google's Mobility Report for Spain, which expresses time spent in these locations in percentage change terms using mobile phone location data. Figure 6a plots the two series, which track each other closely throughout the sample, albeit with more weekly seasonality in the card spending data. In the overall sample of days reported in figure 6a, the correlation is 0.94.

### 2.3.2. Income and mobility

Previous literature has highlighted that lower-income workers are more likely to have jobs for which working from home is not possible [20], but whether such workers continue to work, or suspend their labour market activity and remain at home, is not clear. Figure 6b plots the change in transportation spending during lockdown among cardholders residing in the lowest- and highest-decile Madrid postal codes (by income *per capita*). The average spending reduction relative to pre-COVID baseline for the former is 66% and for the latter is 85%, which is the maximum average reduction for any postal code decile (see [16] for evidence on mobility by postal code in New York City that comes from mobile phones). Strikingly, these differences emerge primarily during the workweek: transport spending falls across postal codes appear much more similar during weekends than those during working days. This strongly suggests that mobility differences across income groups arise because of different work patterns, not because of an innate preference for travel by lower-income households. It also suggests that a substantial number of workers unable to work from home continue to work in lockdown, even if in theory only essential workers were supposed to leave home.

To further explore the relationship between income and mobility during lockdown, we look at spending patterns by Madrid postal code during the peak of the lockdown in April 2020. Figure 7a plots the share of online spending in total spending during this period against postal code income *per*

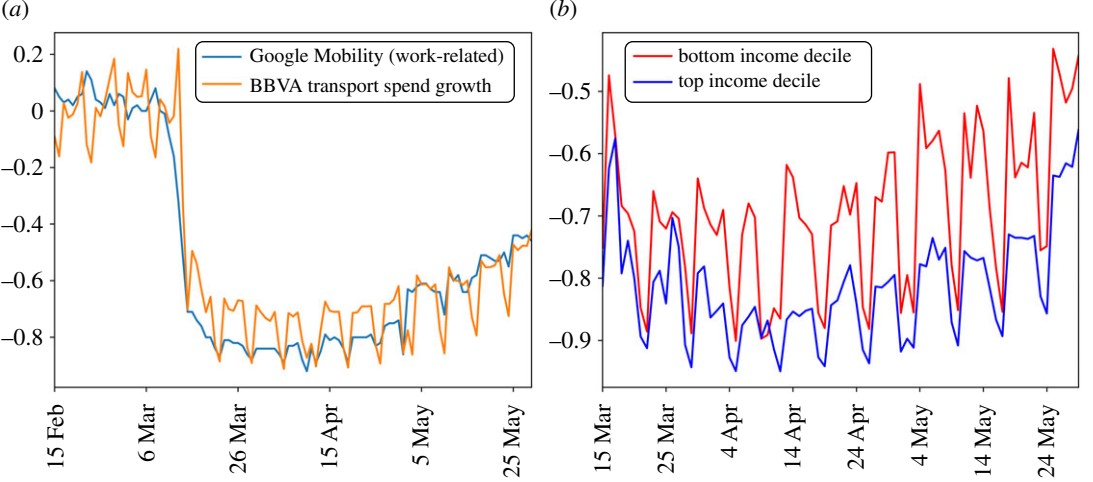

**Figure 6.** (*a*) Comparison of Google Mobility Report for Spain for work-related categories against BBVA card data spending on transportation subcategories. The baseline for computing growth for the BBVA series is the spending average from 1 January to 14 February 2020. (*b*) Change in transport spending among top-income and bottom-income Madrid postal codes during lockdown period.

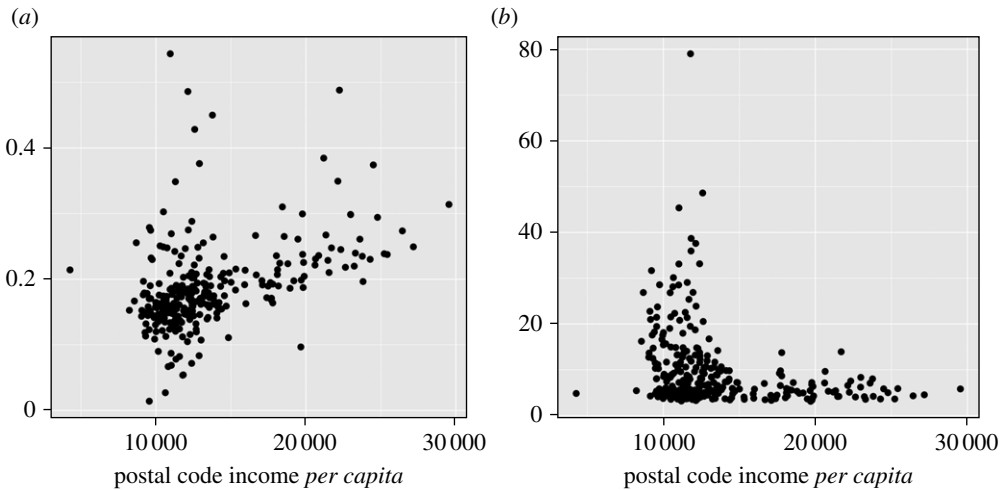

**Figure 7.** This figure compares shopping behaviour related to mobility across Madrid postal codes during April 2020. (*a*) Share of total spending in April 2020 purchased online. (*b*) For each postal code, we estimate the distance travelled in kilometres for making offline purchases in April 2020.

*capita*.[4] The raw correlation between the variables is 0.43 (*p*-value $< 1 \times 10^{-13}$), although the plot makes clear there is substantial variation in online shopping behaviour across all income groups. This nevertheless provides evidence that residents of higher-income postal codes are more able to shop online and avoid leaving their homes during lockdown.

We next examine the distance travelled across postal codes conditional on making offline purchases, which by definition requires leaving one's home. To do this, we first create a dyadic dataset in which we tabulate the offline purchases made by residents of each postal code in all other postal codes in Madrid (including one's own postal code). $share_{ij}$ is the share of offline spending[5] of postal code $i$ purchased in postal code $j$. We then compute $d_{ij}$, the distance in kilometres between geographical centroids of postal

---

[4]In the electronic supplementary material, we show that richer neighbourhoods also had a substantial *increase* in online spending of food (a necessity good) during the pandemic.

[5]In the electronic supplementary material, we further extend the analysis counting the number of transactions (instead of the share of purchases). We show that during the lockdown the number of offline transactions performed outside their postal code by residents of richer neighbourhoods felt much more than in poorer ones, while there are no substantial changes within the neighbourhood of residence.

codes $i$ and $j$. Finally, our estimate of the distance travelled for offline shopping of postal code $i$ residents is $\sum_{j \neq i} d_{ij}$ share$_{ij}$. That is, we weight the physical distance between postal codes by spending shares, and impute a zero distance to purchases made in own postal code. Figure 7$b$ plots this estimate against postal code income *per capita*.[6] Among postal codes with income *per capita* above 15 000, the average distance travelled for offline purchases is 5.7 km and the interquartile range is (4.2 km, 6.6 km). Among postal codes with income *per capita* below 15 000, the corresponding statistics are 11.2 km and (5.2 km, 13.3 km). The implication is that not only are residents of poorer postal codes less likely to make purchases online, but also more likely to travel greater distances when they leave home to make offline purchases. Both facts combine to provide further support to the idea that substantial mobility inequality existed across income groups during Spain's lockdown. Moreover, this demonstrates that card purchase data can be informative about physical movements across narrow geographical units.

### 2.3.3. The infection cost of mobility

A natural next question is whether mobility has health consequences. To the extent that travel outside the home makes it more likely to interact with others, it may increase the risk of contracting coronavirus. Our results above motivate us to use card spending on transportation as an input into a disease model to explore this connection. Furthermore, we explore how different modes of transportation affect disease incidence. From 1 February to 30 April 2020, two modes of transport make up 75% of total spending in Madrid postal codes on transportation: gasoline (63% of total spending) and urban transport (12%). We take the former as a proxy for car transportation, while the latter represents spending on Madrid's public transportation system. A reasonable expectation is that public transportation brings travellers into closer contact with others, so might represent a particularly high-risk form of mobility during the pandemic. This represents another application of card data as a consumption survey, as the detail provided by the spending categories allows us to dig into impacts of different types of travel.

To begin the analysis, we regress total COVID-19 cases per 1000 residents in each postal code on income *per capita* (measured in units of 1000 EUR), the share of residents above 65, and total spending *per capita* on transportation of different forms during February, March and April 2020. The estimated coefficients are in table 4. As expected, the share of older residents is a strong predictor of total cases but we find no effect of income *per capita*. More pertinent for our purposes, we find a moderately strong impact of total transportation spending on cases. The estimated coefficient implies that 1 s.d. change in total transport spending generates a 0.143 s.d. change in COVID-19 cases. This is consistent with transport spending correlating with social contact and disease exposure, which thereby increase disease incidence.

We also find strong heterogeneity in the association between types of transport spending and disease. Spending on car transportation has no significant effects on COVID-19 incidence, but spending on urban transport has very strong effects. The estimated coefficient implies that a 1 s.d. change in urban transport spending generates a 0.267 s.d. change in COVID-19 cases, nearly twice the effect of generic spending. The final column pools urban and car spending. As expected, given car spending makes up most of this combined category, the effects are quite similar to car spending alone.

This highlights that the mode of transportation may be as big a component of health risk as mobility *per se*. Prior to lockdown, higher-income neighbourhoods have a slightly higher share of urban transport spending in total transport spending than lower-income ones. During lockdown, urban transport shares are uncorrelated with income at the postal code level.

There are many factors that the cross-sectional regressions do not control for. Distance from the centre of Madrid, occupational structure, quality of housing stock and population density are all factors that might potentially drive the relationship between disease and mobility. To address these sources of confounding, we next adopt a panel regression framework that allows us to study the impact of transportation spending at daily frequency on disease outcomes *within* postal codes while controlling for postal code fixed effects. The Methods section below formally describes the Poisson regression model we adopt.

Table 5 reports the estimated coefficients of the panel model. As expected, we find significant and positive effects of the lockdown on case growth (since COVID-19 cases peaked during this time) as well as of lagged new cases (since infection dynamics are persistent). All lagged transport spending indicators are positive and highly significant, including car transportation. The interpretation is that

---

[6]There exists a literature aiming to understand distance to the consumption point. See for instance [21]

**Table 4.** Estimated coefficients of ordinary least-squares model for total cases per 1000 residents at Madrid postal code level. Standard errors in parentheses.

| cumulative COVID-19 incidence within postal code | | | | |
|---|---|---|---|---|
| | (1) | (2) | (3) | (4) |
| total transport spending | 0.472** | | | |
| | (0.027) | | | |
| car transport spending | | 0.055 | | |
| | | (0.039) | | |
| urban transport spending | | | 0.760*** | |
| | | | (0.201) | |
| urban + car spending | | | | 0.070* |
| | | | | (0.036) |
| income *per capita* | −0.008 | 0.027 | −0.039 | 0.018 |
| | (0.047) | (0.043) | (0.046) | (0.044) |
| senior share | 30.872*** | 32.372*** | 23.588*** | 31.729*** |
| | (3.576) | (3.591) | (4.124) | (3.564) |
| $R^2$ | 0.272 | 0.261 | 0.297 | 0.267 |
| N | 248 | 248 | 248 | 248 |

***$p < 0.01$; **$p < 0.05$; *$p < 0.1$.

there is a robust relationship between current disease incidence in postal codes, and transportation spending across all categories several weeks prior. Again, though, the effect of urban transport spending is particularly high. In the Poisson model, the average treatment effect is the estimated coefficient value multiplied by the mean of the dependent variable, in this case 3.92. In these terms, a 1000 EUR unit increase in lagged urban transport spending increases daily incidence by 0.49, while the corresponding numbers for overall and car spending are 0.14 and 0.27, respectively.[7]

Overall, then, we observe that transport spending is a good proxy of mobility, as well as a predictor of disease. Since we also observe that residents of poorer postal codes travel more during the workweek in lockdown, the overall implication is that they are also more subject to disease risk than residents of richer neighbourhoods. This is another sense in which card data helps uncover the distributional impact of COVID-19, in this case on expected health outcomes instead of consumption behaviour.

## 3. Discussion

The increasing abundance of detailed and granular financial transactions stored by banks and payment systems is potentially transformative for economic measurement. National statistics agencies are at the earliest stages of engaging with non-traditional data, and our results suggest the value in complementing traditional surveys with naturally occurring transaction data. These efforts are particularly important in low- and middle-income countries, where more standard high-quality and high-frequency indicators of consumption may be too costly to produce.

Transaction data also provide timely signals to policymakers about the impact of economic shocks and policy interventions, which is especially important at times of high uncertainty and rapid change as during the current COVID-19 crisis. We draw three lessons from the first Spanish lockdown in early 2020 from BBVA card data that are more broadly relevant as many countries in the world again limit economic activity to control disease spread.

First, the closing and opening of establishments had a dramatic effect on spending, which reacts abruptly to both measures. On the other hand, social distancing policies and restrictions of capacity

---

[7]We do not observe whether a unit of urban transport spending generates more or less movement through space than a unit of gasoline spending, which would also be an important input into a model of disease risk and spending.

**Table 5.** Estimated coefficients of Poisson regression model for postal-code level COVID-19 incidence. Standard errors in parentheses. The *p*-values associated with each coefficient are less than $1 \times 10^{-15}$.

| daily COVID-19 incidence within postal code | | | | |
|---|---|---|---|---|
| | (1) | (2) | (3) | (4) |
| lagged total transport spending | 0.036*** | | | |
| | (0.0004) | | | |
| lagged total car spending | | 0.070*** | | |
| | | (0.0007) | | |
| lagged urban transport spending | | | 0.125*** | |
| | | | (0.0021) | |
| lagged urban transport + car spending | | | | 0.051*** |
| | | | | (0.0006) |
| lockdown indicator | 1.637*** | 1.658*** | 1.461*** | 1.634*** |
| | (0.0178) | (0.0177) | (0.0177) | (0.0177) |
| lagged daily incidence | 0.023*** | 0.023*** | 0.026*** | 0.028*** |
| | (0.0002) | (0.0002) | (0.0002) | (0.0002) |
| postal code F.E. | Y | Y | Y | Y |
| N | 26 784 | 26 784 | 26 784 | 26 784 |

have a much more limited effect. This highlights that lockdown policies are not an either/or policy. When countries ease out of lockdown, shop openings are important for stimulating economic activity, but capacity restrictions can be maintained for longer periods at relatively low economic cost while protecting health.

Second, underlying this decline in expenditure is a large reallocation across expenditure categories, away from social goods and luxuries. As a result, higher-income groups—those who consume such goods relatively more in normal times—saw their spending decline by more. The resulting increased savings suggests that private households in high-income neighbourhoods accumulated assets during the crisis that could help finance the large government deficits resulting from employment support and other measures.

Third, detailed transaction data on transportation and commuting expenditures reveals that residents of poorer neighbourhoods are more likely to travel during the workweek during lockdowns, and that this correlates with higher disease incidence. Importantly, though, the mode of transportation appears to affect disease. Investment in additional safety measures for users of public transportation, and in transportation infrastructure that promotes social distancing without increasing pollution (e.g. cycle lanes), could mitigate these impacts.

Overall, our paper demonstrates how transaction data can be used to assess economic conditions. We show that such data are able to capture many relevant patterns in spending and that, importantly, it does so in near-real time. Moreover, its unprecedented granularity offers the possibility of using it as a high-resolution *microscope*; not only for deciding how best to weather future shocks—pandemic-related or otherwise—but also to provide the tools for an ever more granular and covariate-rich analysis of both economic events and economic models.

# 4. Methods

## 4.1. Transaction data

The bulk of our analysis centres on Spanish transaction data. Our data for Spain consist of a join between (a) the universe of transactions at BBVA-operated point-of-sale (PoS) and (b) the universe of transactions by BBVA-issued credit and debit cards (in non-BBVA-owned PoS, to avoid double counting). The time

stamps of transactions available to us range from 1 January 2019 till 29 June 2020. All data were anonymized prior to treatment and aggregated at BBVA before being shared externally.

In the electronic supplementary material, we present some summary statistics of this large dataset. In total, our analysis builds up from 2.1 billion card transactions, with about two-thirds of the observations in 2019 and the remainder in 2020. At one end of each transaction is a PoS. We observe 2 (1.6) million distinct PoS in 2019 (2020, respectively). The median transaction in either year is just under 20 EUR, with the overall distribution of transactions spanning three orders of magnitude, from 2 EUR to 200 EUR at the 5th and 95th percentile of transaction values.

Each transaction is tagged with information on whether it was carried out at an online PoS (e.g. an internet purchase) versus offline, at a physical PoS. In this data, 30% of all 2019 PoS are online, accounting for 8.4% of all transactions. Note that all online transactions are necessarily completed with a debit or credit card while offline transactions can occur via either card (which we observe) or cash (which we do not). This implies that our sample of expenditures is biased towards online expenditures.

Furthermore, for each PoS, we have a classification of the principal activity of the firm selling goods and services through that PoS. This classification breaks down the universe of transactions into 76 categories, ranging from toy stores to funeral homes.

We are also able to distinguish whether the card initiating each transaction was issued by a Spanish bank or by a foreign bank. Throughout, we mainly focus on national card transactions, which account for 93% of the transactions in the sample. Within the sample of national card transactions we sometimes focus on the subsample of BBVA cardholders. In 2019, there are 6.3 million unique BBVA cardholders. This comprises a 16% sample of Spain's adult population of 39 million.

For BBVA cardholders, we observe their home address postal code, their education level and age. In the electronic supplementary material, we compare the age structure and educational attainment of BBVA cardholders to that of Spain's adult population. Overall, our sample is broadly in line with the latter on both dimensions, somewhat undersampling the youngest and oldest in the population while oversampling the middle aged.

When analysing these data, we calculate Y-o-Y growth as follows: we pair every day following 8 January 2020 with its equivalent weekday in the equivalent week of the previous year. Thus, given that Epiphany is one of the most important holidays of the year in Spain and we exclude Y-o-Y comparison over the holiday period, we pair the first Tuesday after the Epiphany holiday in 2020 (8 January) with the first Tuesday after Epiphany in 2019 (7 January), and we then proceed daily, always pairing days of the week (first Wednesday with first Wednesday, etc.). We then measure the 2019–2020 Y-o-Y growth for the same day of the week. This controls for weekly seasonality to some extent, but to further control for weekly variation in some of the graphs we use the 7-day moving average. In figure 2, it is particularly important to control for day of the week variation, so we show the residuals on day of the week dummies.

Finally, note that expenditures are measured in nominal terms throughout and our data does not include any price-level information. Particularly for our COVID-19 applications, note that it is likely that the relevant deflators are changing substantially as the crisis unfolds.

## 4.2. Postal-code level data

To obtain a measure of income at the postal code level, we build up from a granular cross-section of data available from the Spanish Statistical Office (INE) referring to 'secciones censales'. These are small spatial divisions (equivalent to US Census tracts) and homogeneous in size, forming groups of around 1500 individuals each. For each of these groups, we know their aggregate taxable income (from tax returns of residents in each 'sección censal').

The Health authorities of the Autonomous Community of Madrid divide the region in 286 health districts of approximately uniform size as their basic unit for the provision of health services, and they report the daily incidence of the pandemic in each of those districts.

To account for the differential incidence of the pandemic across the geography of Madrid, we use the geographical position of health districts and postal codes to calculate and impute the daily incidence of confirmed COVID-19 cases within the different postal codes.

There are some technical caveats. We have information on disease incidence for health districts, while we have information on expenses from BBVA by postal code, and we have socio-economic information at 'sección censal' level. Unfortunately, the three levels do not have a perfect match, but we have detailed geo-location information of the three levels, so we can place them in the map exactly. To merge the three sources of data, we have used the following procedure:

(i) The smallest in size of the three units is by far the 'sección censal', which consists of very homogeneous divisions of around 1500 individuals. Postal codes and health districts are larger, and of comparable sizes.

(ii) We calculate the socio-economic status of each postal code by merging the information of all the 'secciones censales' that are completely included within the postal code.

(iii) In order to attribute COVID-19 incidence to each postal code, we assume that incidence is uniformly distributed across the inhabitants of the specific health district, and impute to each 'sección censal' within the health district its proportional share. We then sum the imputed COVID-19 incidence of the 'secciones censales' that are within each postal code to determine the degree of incidence within it.

An additional issue is that the reported number is not the daily incidence, but the accumulated one for the previous 14 days (or aggregated) and there seem to be revisions of the data when cases are diagnosed incorrectly, etc. We calculate daily incidence as the difference between the reported accumulated incidence one day and the one reported the previous day.

## 4.3. Further analysis of lockdown easing

The standard difference-in-differences (DD) analysis for the effect of lockdown easing exploits variation across groups of provinces that receive treatment (i.e. lockdown easings) at different times. One first concern that arises is that different provinces were on different pre-treatment expenditure trends. We address this concern by focusing on the differential effects of Phase 1 easing, the largest point estimate obtained. Specifically, there are two groups of provinces that are of interest: the early easers, switching to Phase 1 on 11 May versus later easers, coming out of Phase 0 only in the subsequent weeks. We start by noting that, pre-8 March, there is no statistically significant differential trend in expenditures across these two groups of provinces. Furthermore, the same conclusions arise when looking at the differential expenditure trends within the lockdown period or within the Phase 0 period, when both sets of provinces were subject to the same nationwide restrictions. Early switchers' daily growth during the pre-lockdown period is, on average, 1.8 percentage points higher than that of late switchers but the associated $p$-value is 0.195. Alternatively, taking the first 10 days of May as the relevant pre-treatment period gives an insignificant 0.01 percentage point difference. Conclusions are unchanged by defining different pre-treatment periods within the joint lockdown and Phase 0 periods.

A second concern that arises, as articulated in [22], is that the treatment effect may not be stable over time. In our context, this means that the expenditure effects of lockdown easing may be different across early- and late-switcher provinces, perhaps indicating that other unobservable time-varying factors are driving the province-level response. To address this concern, we again focus on Phase 1 treatment effects. To do this, we zoom in on the period running through 25 May, when all Spanish provinces remained in either Phase 0 or 1. Thus, within this subsample, we have three groups of provinces: early switchers, easing into Phase 1 on 11 May, late switchers on 18 May and never switchers (till 25 May). Based on this classification, we can use the [22] decomposition theorem to estimate changes in Phase 1 treatment effects across different subgroups. Our estimates imply stable treatment effects. The DD estimate based on the difference between early and late switchers is 0.157. The converse estimate based on effects on late switchers versus those that had already eased previously, gives a DD estimate of 0.139. Finally, the DD estimate formed by the differential growth between ever treated and never treated gives 0.153. We conclude that, at least for the case of Phase 1, the treatment effect is stable with respect to the timing of treatment.

## 4.4. Poisson regression model for disease outcomes as function of spending

Let $y_{i,t}$ be the number of new COVID cases in postal code $i$ on day $t$, and $x_{i,t}$ be the level of transport spending of postal code $i$ resident on day $t$, measured in 1000 EUR units. The within-postal-code disease predictor we use is $x_{i,(t-28):(t-14)} = \frac{1}{14}\sum_{\tau=t-28}^{t-14} x_{i,\tau}$, which accounts for two aspects of transport spending. First, it is potentially noisy, so averaging over multiple days helps dampen the impact of idiosyncratic, day-level spending variation. Second, it accounts for the incubation time of coronavirus before the onset of COVID-19, as well as delays in testing and the recording of cases in official statistics. Our construction focuses on the health impact of transport spending on a given day on

disease outcomes two-to-four weeks later. Averaging also helps control for the uncertainty in the exact timing of the health effects.

We model $y_{i,t}$ using a Poisson regression model[8] with mean

$$\mu_{i,t} = \beta_1 y_{i,t-1} + \beta_2 x_{i,(t-28):(t-14)} + \beta_3 \text{Lockdown}_t + \gamma_i.$$

$\text{Lockdown}_t$ is an indicator variable for whether a day falls in the post-lockdown period (recall that our case data begin in late February prior to lockdown) and $\gamma_i$ is a postal code fixed effect which controls for any time-invariant postal-code characteristics that might affect disease outcomes or transport spending.

Data accessibility. In the electronic supplementary material, we make available codes and data—both expenditure series and necessary covariates—pertaining to the national-, province- and broad category-level data, allowing researchers to fully replicate key COVID-19 results in the paper (figures 1, 2, 4 and table 1) and to conduct their own national, subnational or expenditure-category analysis in this context. Note, however, that this study builds from proprietary card transaction data from BBVA, a Spanish commercial bank. Both the individual-level card source data and aggregations at the postal code level or highly disaggregated category level involve highly sensitive personal information about customers and/or may disclose proprietary commercial information on local bank activities. Therefore, we are unable to render data fully publicly accessible beyond what is deposited in the electronic supplementary material. In particular, we are unable to publicly share replication materials involving: historical time series for Spain, cross-country expenditure data, detailed category of expenditure information or postal code level data. Individual researchers interested in these more detailed datasets should direct their query to BBVA Research. The Spanish Household Budget Survey is publicly accessible data, and can be obtained from the web page of the 'Instituto Nacional de Estadística' The data on income at census tract level (CUSEC) from where the income at postal code level is calculated is also public, and can also be obtained from https://www.ine.es/experimental/atlas/exp_atlas_tab.htm the web page of the 'Instituto Nacional de Estadística' The data on incidence of the pandemic at Madrid Health District level can be obtained from: https://www.comunidad.madrid/servicios/salud/2019-nuevo-coronavirus.

Authors' contributions. V.M.C., S.H. and J.V.R.M. analysed the data and wrote the manuscript. J.R.G., A.O., T.R. and P.R. analysed the data.

Competing interests. We declare we have no competing interests.

Funding. V.M.C. gratefully acknowledges funding from the Leverhulme Trust and the European Research Council, grant no. 101001221, MICRO2MACRO. S.H. gratefully acknowledges funding from the European Research Council, grant no. 864863. J.V.R.M. gratefully acknowledges the support of the UK Economic and Social Research Council (ESRC), award reference ES/L009633/1.

Acknowledgements. The authors thank four anonymous reviewers and editors for their valuable suggestions. We also thank Raman Singh Chhina for superb research assistance, and Gary Koop for inspiring comments on a previous draft.

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
