## [Peer Review File · Royal Society Open Science]

Review History

RSOS-210218.R0 (Original submission)

Review form: Reviewer 1

Is the manuscript scientifically sound in its present form?

Yes

Are the interpretations and conclusions justified by the results?

Yes

Is the language acceptable?

Yes

Do you have any ethical concerns with this paper?

No

Have you any concerns about statistical analyses in this paper?

No

Recommendation?

Accept with minor revision (please list in comments)

Comments to the Author(s)

This is a great contribution, and I would probably publish as is. I'll offer three substantive comments, though.

(1) My main reservation is that I felt that this was basically two papers: one on data validation and another about COVID. Personally, I was very excited about validation of the BBVA data (I am tired of COVID), and I was disappointed that it felt like it got short shrift by relegation to Section 2 of the Appendix. The abstract promised "extensive validations exercises," and I was hoping for more. Hopefully, the lessons of covid will grow less relevant, and the BBVA (and other transaction) data will grow more prevalent!

(2) Section 2.3 was my favorite part of the paper, and it did feel like there were a few opportunities for improvements, here.

- Rich people spend more online (Fig. 7a) -- I would love to know if they were buying more expensive stuff online, or shielding themselves from exposure for necessary transactions. (Several very old friends in Spain were doing this, but I don't know if it was as common as in the US.) Even if local supermarkets etc. were handling grocery delivery, and not big online firms,, presumably you can still use whether or not the card was present at the sale (that is usually in the credit card data).

- Fig 7b: There is a lot that could be said about distances traveled for essential transactions, aka food deserts and amenities, and the authors should probably look up that literature some.

- Distances traveled. This is where I hoped this paper was going, and it could be sharpened a bit. I was not convinced by the use of share-weighted formula (authors' line 315), in defining the distances traveled. In short, I would have summed the total activity (# of transactions) weighted PERHAPS but not necessarily by distance. If I am constructing total amount of mobility, distance is not based on shares, it is based on the number of transactions. (Caveat: of course, people string together routines, but let's not get carried away...) Just for illustration, if I am very stubborn and I insist on an espresso out, every morning, I am much more exposed than by buying a big TV to get me through the pandemic. Not only does this allow large purchase to dominate unnaturally (the location of big purchases matters more), but I think it misses the point of exposure, which is basically one per interaction. You may disagree, but then please state why.

- The analysis of public and private transportation was particularly interesting, but I did not find it intuitive, mainly because the costs are different. I have no idea what 1000 EUR of car travel vs bus travel buys (line 368). You quote car as "gasoline" (not parking, and the vehicles are outside the data), so is each trip... cheaper?

- Can you see substitution from public to private transport, perhaps more extreme in rich areas? If you have individual cardholders, you could do this at a very granular level!

- I would add another column to Tables 4 and 5 with both public transport and car spending.

(3) Appendix Fig 2b: Yes, the correlations are very high, but the slope is NOT $x = y$. BBVA share is depressed at low INE share, suggesting that poorer populations do less of their spending through BBVA. Is that just because housing and fixed costs via bank transfers are a larger share? Or is it because they use cash more and/or have less access to financial services like BBVA? What does this slope imply for your estimates? Can you estimate if this difference is all "taken up" in the rents and durables? (cf app. lines 22-29).

The balance of my comments are quibbles and typos, which hopefully nevertheless improve the polish.

Typos, quibbles, etc.

(All line numbers are the authors' rather than RSOS's)

30: citation weirdness on Katherine Abraham (K, other authors missing etc.)

43: "data consists OF the universe"

63 and 268: I found it a little impolitic to use the word "suffer" for larger expenditure drops by the wealthy, during a pandemic.

75-93: a bunch of `\citep{}` vs `\cite{}` weirdness going on. If you're using the authors names in a sentence, it shouldn't be in parentheses.

87: "Consistent with this, we find local disease incidence to be a driver of expenditure *growth*, ..." > is the sign flipped? i.e., disease incidence *depresses* expenditures rather enhancing it.

137: your call, but I think the word "data" is plural: "data end, ..."

fig. 1: vertical lines could be annotated much more clearly in-plot.

fig. 3 and 212: correlation coefficient 0.865 -- what is it, without groceries & the most-extreme outliers?

295: Missing space before "(See..."

457, 460: "secciones censales" fix the flat quote marks

478: "uniformly distributed across the specific health district" Uniform with respect to what? Land area? People? (I hope!)

Table A1: I understand that this is population 18+, but in panel B, could make it age 18-25 -- would be more clear than < 25.

Review form: Reviewer 2

Is the manuscript scientifically sound in its present form?

No

Are the interpretations and conclusions justified by the results?

Yes

Is the language acceptable?

Yes

Do you have any ethical concerns with this paper?

No

Have you any concerns about statistical analyses in this paper?

No

Recommendation?

Major revision is needed (please make suggestions in comments)

Comments to the Author(s)

There is a lot of rich information in this paper. Thank you. As I understand it, there are two intended contributions in this paper:

1. Show the advantages of transaction-level data over survey data for economic measurements, including national accounting.
2. Provide some empirical evidence on the effects of COVID and the lockdowns.

To summarize, I do not see the COVID part of the paper as a contribution other than to a COVID-specific issue. As for the methodological part, I can see how that can turn into a valuable contribution for the scientific community, including the supplementary material in this paper, provided the underlying data (not the micro data, but the province-level or group level, aggregated data) is released in a clean, transparent way.

Some more detailed comments:

Regarding (1), I find the context of COVID insufficient. During COVID, a lot of transactions were moved online, minimizing the common challenges of capturing a representative sample of economic activity with just one bank's information. I agree that (1) would be an important contribution, not only to Economics, but to the whole scientific community. But I would not mix it with anything regarding COVID and I'd expand the scope to the years before the pandemic, to show that these results hold more generally also when people pay in cash more often. Including the material in the Supplementary Appendix and some of the "Validation" parts in the main text.

In this case, the data release in its current form is not sufficient. Transparency in measurement is a sufficient contribution, but it has to come with transparency. I apologize in advance if this is on my end, but the data in the link (<https://www.bbvarresearch.com/en/special-section/charts>) contains only a subset of the data used in the paper. In addition, I found no way to download the data behind the graphs. Data should be provided in (csv, excel, txt...) tables to help future research. IF this is indeed a problem on my end, that I did not find a way but it is there, please consider making the actual data easier to find/download.

While I understand the limitations of disclosing the underlying data, I did not see any microdata involved in the main results paper in a way that privacy would be compromised. If there is to be a paper about the measurement advantages of banking data, aggregated and anonymized data that meets usual requirements to protect privacy, fully available and not only "upon request", should be released in a clean replication package.

To point (2), the evidence presented in this paper contains some interesting and novel facts regarding behaviour of individuals around the lockdowns and pandemic, the dynamics and heterogeneity of consumption during the lockdowns, and on mobility. The different pieces of evidence, however, are relatively unstructured. I would rather focus on the methodological contribution and send the COVID part to a COVID-specific empirical or policy issue, such as <https://cepr.org/content/covid-economics-vetted-and-real-time-papers-0>.

I understand this might be too big of a change, but I recommend "major revision" in the hope that the methodological part can turn into a clean, transparent, tool for consumption measurement that would indeed be of great value to the scientific community.

Review form: Reviewer 3

Is the manuscript scientifically sound in its present form?

Yes

Are the interpretations and conclusions justified by the results?

Yes

Is the language acceptable?

Yes

Do you have any ethical concerns with this paper?

No

Have you any concerns about statistical analyses in this paper?

No

Recommendation?

Accept with minor revision (please list in comments)

Comments to the Author(s)

Please find them in my report (see Appendix A).

Review form: Reviewer 4

Is the manuscript scientifically sound in its present form?

Yes

Are the interpretations and conclusions justified by the results?

Yes

Is the language acceptable?

Yes

Do you have any ethical concerns with this paper?

No

Have you any concerns about statistical analyses in this paper?

Yes

Recommendation?

Accept with minor revision (please list in comments)

Comments to the Author(s)

Attached file contains report (see Appendix B).

Decision letter (RSOS-210218.R0)

Dear Professor Carvalho

On behalf of the Editors, we are pleased to inform you that your Manuscript RSOS-210218 "Tracking the COVID-19 Crisis with High Resolution Transaction Data" has been accepted for publication in Royal Society Open Science subject to minor revision in accordance with the referees' reports. Please find the referees' comments along with any feedback from the Editors below my signature.

Please submit your revised manuscript and required files (see below) no later than 7 days from today's (ie 09-Jun-2021) date. Note: the ScholarOne system will 'lock' if submission of the revision is attempted 7 or more days after the deadline. If you do not think you will be able to meet this deadline please contact the editorial office immediately.

on behalf of Marta Kwiatkowska (Subject Editor)
openscience@royalsociety.org

Associate Editor Comments to Author:
Comments to the Author:

Thank you for your patience while we sought reviewers of your work - unfortunately, the continuing disruptions wrought by COVID and the pressures on many experts to review a rising tide of COVID literature has slowed the journal more than we (and no doubt you) would have preferred. Nevertheless, I'm pleased that we've received a number of reports and all are broadly positively inclined towards your work. However, a number of comments have been raised by each reviewer and we'd like you to carefully address these - especially those of reviewer 2. Given the volume of commentary, if you need a slight extension on your revision deadline, please contact the editorial office for advice/assistance. Good luck and we'll look forward to reading your revision in the near future.

Reviewer comments to Author:
Reviewer: 1

Comments to the Author(s)
This is a great contribution, and I would probably publish as is. I'll offer three substantive comments, though.

(1) My main reservation is that I felt that this was basically two papers: one on data validation and another about COVID. Personally, I was very excited about validation of the BBVA data (I am tired of COVID), and I was disappointed that it felt like it got short shrift by relegation to Section 2 of the Appendix. The abstract promised "extensive validations exercises," and I was

hoping for more. Hopefully, the lessons of covid will grow less relevant, and the BBVA (and other transaction) data will grow more prevalent!

(2) Section 2.3 was my favorite part of the paper, and it did feel like there were a few opportunities for improvements, here.

- Rich people spend more online (Fig. 7a) -- I would love to know if they were buying more expensive stuff online, or shielding themselves from exposure for necessary transactions. (Several very old friends in Spain were doing this, but I don't know if it was as common as in the US.) Even if local supermarkets etc. were handling grocery delivery, and not big online firms,, presumably you can still use whether or not the card was present at the sale (that is usually in the credit card data).

- Fig 7b: There is a lot that could be said about distances traveled for essential transactions, aka food deserts and amenities, and the authors should probably look up that literature some.

- Distances traveled. This is where I hoped this paper was going, and it could be sharpened a bit. I was not convinced by the use of share-weighted formula (authors' line 315), in defining the distances traveled. In short, I would have summed the total activity (# of transactions) weighted PERHAPS but not necessarily by distance. If I am constructing total amount of mobility, distance is not based on shares, it is based on the number of transactions. (Caveat: of course, people string together routines, but let's not get carried away...) Just for illustration, if I am very stubborn and I insist on an espresso out, every morning, I am much more exposed than by buying a big TV to get me through the pandemic. Not only does this allow large purchase to dominate unnaturally (the location of big purchases matters more), but I think it misses the point of exposure, which is basically one per interaction. You may disagree, but then please state why.

- The analysis of public and private transportation was particularly interesting, but I did not find it intuitive, mainly because the costs are different. I have no idea what 1000 EUR of car travel vs bus travel buys (line 368). You quote car as "gasoline" (not parking, and the vehicles are outside the data), so is each trip... cheaper?

- Can you see substitution from public to private transport, perhaps more extreme in rich areas? If you have individual cardholders, you could do this at a very granular level!

- I would add another column to Tables 4 and 5 with both public transport and car spending.

(3) Appendix Fig 2b: Yes, the correlations are very high, but the slope is NOT $x = y$. BBVA share is depressed at low INE share, suggesting that poorer populations do less of their spending through BBVA. Is that just because housing and fixed costs via bank transfers are a larger share? Or is it because they use cash more and/or have less access to financial services like BBVA? What does this slope imply for your estimates? Can you estimate if this difference is all "taken up" in the rents and durables? (cf app. lines 22-29).

The balance of my comments are quibbles and typos, which hopefully nevertheless improve the polish.

Typos, quibbles, etc.

(All line numbers are the authors' rather than RSOS's)

30: citation weirdness on Katherine Abraham (K, other authors missing etc.)

43: "data consists OF the universe"

63 and 268: I found it a little impolitic to use the word "suffer" for larger expenditure drops by the wealthy, during a pandemic.

75-93: a bunch of `\citep{} vs \cite{} weirdness` going on. If you're using the authors names in a sentence, it shouldn't be in parentheses.

87: "Consistent with this, we find local disease incidence to be a driver of expenditure *growth*, ..." > is the sign flipped? i.e., disease incidence *depresses* expenditures rather enhancing it.

137: your call, but I think the word "data" is plural: "data end, ..."
 fig. 1: vertical lines could be annotated much more clearly in-plot.
 fig. 3 and 212: correlation coefficient 0.865 -- what is it, without groceries & the most-extreme outliers?
 295: Missing space before "(See..."
 457, 460: "secciones censales" fix the flat quote marks
 478: "uniformly distributed across the specific health district" Uniform with respect to what? Land area? People? (I hope!)

Table A1: I understand that this is population 18+, but in panel B, could make it age 18-25 -- would be more clear than < 25.

Reviewer: 2

Comments to the Author(s)

There is a lot of rich information in this paper. Thank you. As I understand it, there are two intended contributions in this paper:

1. Show the advantages of transaction-level data over survey data for economic measurements, including national accounting.
2. Provide some empirical evidence on the effects of COVID and the lockdowns.

To summarize, I do not see the COVID part of the paper as a contribution other than to a COVID-specific issue. As for the methodological part, I can see how that can turn into a valuable contribution for the scientific community, including the supplementary material in this paper, provided the underlying data (not the micro data, but the province-level or group level, aggregated data) is released in a clean, transparent way.

Some more detailed comments:

Regarding (1), I find the context of COVID insufficient. During COVID, a lot of transactions were moved online, minimizing the common challenges of capturing a representative sample of economic activity with just one bank's information. I agree that (1) would be an important contribution, not only to Economics, but to the whole scientific community. But I would not mix it with anything regarding COVID and I'd expand the scope to the years before the pandemic, to show that these results hold more generally also when people pay in cash more often. Including the material in the Supplementary Appendix and some of the "Validation" parts in the main text.

In this case, the data release in its current form is not sufficient. Transparency in measurement is a sufficient contribution, but it has to come with transparency. I apologize in advance if this is on my end, but the data in the link (<https://www.bbvaesearch.com/en/special-section/charts>) contains only a subset of the data used in the paper. In addition, I found no way to download the data behind the graphs. Data should be provided in (csv, excel, txt...) tables to help future research. IF this is indeed a problem on my end, that I did not find a way but it is there, please consider making the actual data easier to find/download.

While I understand the limitations of disclosing the underlying data, I did not see any microdata involved in the main results paper in a way that privacy would be compromised. If there is to be a paper about the measurement advantages of banking data, aggregated and anonymized data that meets usual requirements to protect privacy, fully available and not only "upon request", should be released in a clean replication package.

To point (2), the evidence presented in this paper contains some interesting and novel facts regarding behaviour of individuals around the lockdowns and pandemic, the dynamics and heterogeneity of consumption during the lockdowns, and on mobility. The different pieces of evidence, however, are relatively unstructured. I would rather focus on the methodological contribution and send the COVID part to a COVID-specific empirical or policy issue, such as <https://cepr.org/content/covid-economics-vetted-and-real-time-papers-0>.

I understand this might be too big of a change, but I recommend "major revision" in the hope that the methodological part can turn into a clean, transparent, tool for consumption measurement that would indeed be of great value to the scientific community.

Reviewer: 3

Comments to the Author(s)

Please find them in my report. (RSOS-210218_Review.pdf)

Reviewer: 4

Comments to the Author(s)

attached file contains report (report_rsos2021.pdf)

===PREPARING YOUR MANUSCRIPT===

===PREPARING YOUR REVISION IN SCHOLARONE===

Author's Response to Decision Letter for (RSOS-210218.R0)

See Appendix C.

Decision letter (RSOS-210218.R1)

Dear Professor Carvalho,

I am pleased to inform you that your manuscript entitled "Tracking the COVID-19 Crisis with High Resolution Transaction Data" is now accepted for publication in Royal Society Open Science.

COVID-19 rapid publication process:

We are taking steps to expedite the publication of research relevant to the pandemic. If you wish, you can opt to have your paper published as soon as it is ready, rather than waiting for it to be published the scheduled Wednesday.

This means your paper will not be included in the weekly media round-up which the Society sends to journalists ahead of publication. However, it will still appear in the COVID-19 Publishing Collection which journalists will be directed to each week (<https://royalsocietypublishing.org/topic/special-collections/novel-coronavirus-outbreak>).

If you wish to have your paper considered for immediate publication, or to discuss further, please notify openscience_proofs@royalsociety.org and press@royalsociety.org when you respond to this email.

You can expect to receive a proof of your article in the near future. Please contact the editorial office (openscience@royalsociety.org) and the production office (openscience_proofs@royalsociety.org) to let us know if you are likely to be away from e-mail

contact – if you are going to be away, please nominate a co-author (if available) to manage the proofing process, and ensure they are copied into your email to the journal. Due to rapid publication and an extremely tight schedule, if comments are not received, your paper may experience a delay in publication.

on behalf of Marta Kwiatkowska (Subject Editor)
openscience@royalsociety.org

Appendix A

Comments on the RSOS manuscript RSOS-210218: Tracking the COVID-19 Crisis with High-Resolution Transaction Data

The Covid shock has hit economies in a similar way even if its severity differed substantially. Spain is one of the most affected countries so this kind of study that documents the significant changes in consumption patterns is extremely valuable.

There is a lot to like about this paper. The dataset the authors used is very novel, and probably the largest and most comprehensive one that exists in the current literature. By showing simple statistics and regressions, the authors make striking points. I do firmly believe that the paper would make a great contribution to the Royal Society Open Science.

I have some comments below which, I hope, would improve the paper further. I'd like to stress that these are suggestions, rather than firm revision requests. If the authors think the cost of taking these on board (due to, e.g. excessive data processing required) is significant compared to their marginal contribution, a simple discussion related to these points would suffice.

1 Comments

1. How widely is cash used in Spain? Did this change during the pandemic? A simple statistics as to what proportion of the population has debit or credit cards would be useful. Also, the authors can look into excess deposits and check if there was a material change in the money stock during the pandemic.
2. During normal times, what is the share of necessary (that households cannot cut) vs luxurious consumption (that households can cut) for high vs low income households?
3. The authors have information on whether transactions take place online or offline. The natural follow up is to look into how people substituted offline shopping with online. And if cheaper shops became more popular amongst lower income households?

4. It might be an obvious question but debit cards are linked to customers' bank accounts, right? Isn't there a way to get their income data? Excuse the question if it's too obvious or this cannot be done due to data sharing regulations.
5. Do the authors have information as to how big the households are? I.e. who is the main bread maker of the family or if the account is joint?
6. How easy for Spanish households to get a credit card? I.e. Do they have the incentive to use their debit cards more because credit cards are hard to get or are there any incentives (bonus points, cash backs) that make credit card usage more attractive? This is a minor point but credit cards are a way of pushing back household consumption, at least for a month. During the Covid crisis, some households might want to do that more than others especially if they received a big income shock.

2 Minor comments

- Pg 1, footnote 2: After the URL, either 'but' or 'which' is redundant.
- Pg 2, Literature: no need to put all the papers in parentheses.

Appendix B

Referee's report on: Tracking the Covid- 19 crisis with high resolution transaction data; Carvalho et al.

This paper uses individual transaction data from a large Spanish bank to track consumption expenditure through the pandemic, compare the results with traditional survey-type data used by the national statistical agency, look at the effects of lockdown policies, measure mobility and estimate the relationship with external transportation expenditure, and consider differences in behaviour across income groups. The data are exceptionally interesting, the analysis is full of insights and the paper is well written. It is a first-rate piece of work, already well known in this literature, and I certainly recommend publication.

General comment.

1. There is just one general concern that I want to raise, not requiring a modification to the analysis, but rather I would suggest that the authors explain the issue more fully, so that the reader not familiar with the causal inference literature in statistics, or with the problem of endogeneity in economics, will not be misled by some of the results.

I refer primarily to the part of section 2.1 entitled 'Effects of Lockdown and its Easing'. This section uses the language of causation ('effects'), whereas in some other parts of this paper the authors discuss correlation and association. It is of course natural to wish to draw causal inferences about the effects of the lockdown. However, differences in the timing of the easing of restrictions across provinces are presumably not entirely exogenous—as is noted here (as selection into treatment). This nonetheless makes things more complicated than a simple natural experiment with exogenous variation.

Credible attempts are made to deal with this problem, but the simple inclusion of the daily case incidence in Table 1 still leaves us in the realm of conditional expectation rather than causal inference. The difference-in-difference results are a way of attempting estimation of the treatment effect, but the conditions, assumptions and validity checks (although these are further discussed in Section 4) may deserve a more thorough explanation.

It seems unlikely that other standard solutions, for example observation of conditions for a regression discontinuity design, could be put into practice here. Instead, my suggestion to the authors is that they simply describe the issues surrounding estimation of treatment effects more explicitly (ie not assume that the reader is familiar with the econometrics/statistics of causal inference) and acknowledge any limitations which it may imply. In some cases, any bias that is created maybe in the opposite direction to the effect of interest, so that it may even be possible to argue that the observed effect is an underestimate. But in any event, I think that the reader's attention might be drawn more directly to the difficulties involved in causal statistical inference on observational data.

I emphasize that this is an expositional point only.

Minor comments (page references are to the journal's pdf, +2 relative to authors' page numbering)

Page 7, Table 1: Please clarify the precise definitions of the dummy variables used: e.g. the matrix of province dummies has one column for each province, and within each column the variable is 1 up until the date of lockdown easing, 0 on the date of easing and thereafter. Relatedly, page 8, lines 188 and following: in columns 5 and 6 of the table, day fixed effects are added, which changes the omitted category (line 191). The signs as well as magnitudes of coefficients change in Table 1. Interpretation depends on the definitions of dummies; providing an example computation would make this much clearer.

Page 13 line 297: 'during weekends *that* during working days...' → 'during weekends *than* during working days... '

Page 16: The location of the methods section here, after the discussion section (which seems to play the role of conclusion to the paper), seems a little odd. Would it not be sensible to place methods first, wrap the main paper up with the overall discussion and conclusions, and then move to the appendices?

Page 19, discussion of the Poisson regression model. This discussion seems to pertain to the absolute number of new cases, whereas the model earlier in the text used cases per thousand. With the absolute number of new cases, there will of course be more in more highly populated areas, and yet the population of the region does not seem to be used as an explanatory variable (page 19, equation following line 524). If this is the case the omitted population variable would just project onto transportation spending, leading to a biased estimate. Is this just an incomplete description? Please clarify.

Response to Referees and Editors

RSOS Resubmission, ID #RSOS-210218

“Tracking the COVID-19 Crisis with High-Resolution Transaction Data”

Associate Editor Comments

Thank you for your patience while we sought reviewers of your work - unfortunately, the continuing disruptions wrought by COVID and the pressures on many experts to review a rising tide of COVID literature has slowed the journal more than we (and no doubt you) would have preferred. Nevertheless, I'm pleased that we've received a number of reports and all are broadly positively inclined towards your work. However, a number of comments have been raised by each reviewer and we'd like you to carefully address these - especially those of reviewer 2. Given the volume of commentary, if you need a slight extension on your revision deadline, please contact the editorial office for advice/ assistance. Good luck and we'll look forward to reading your revision in the near future.

We thank the Associate Editor for the opportunity to revise our paper and the four reviewers for their extensive comments and suggestions. We were glad to hear that all referees were positively inclined towards our paper.

Below, we provide a point-by-point answer to all questions and comments raised by the reviewers. As a whole, the comments have led to a number of changes and valuable additions to the paper though none of our main qualitative or quantitative results have changed as a result of this revision. While we refer you to the detailed answers below for the specifics, taken together the changes to the paper can be summarized as:

- *Additional results and analysis following on Comment 2 by Reviewer 1 and Comments 2 and 3 by Reviewer 4. These additional results can be found in the Main Paper (Table 4 and 5) and in Supplementary Appendix to the paper.*
- *Clarification of certain points in the analysis, clearer discussions of limitations of our analysis and/ or typographical errors. following on Comments 2, 3 and 4 by Reviewer 1, Main Comment 1 and various Minor Comments by Reviewer 3 and various Minor Comments by Reviewer 4. These additional clarifications, discussions and corrections can be found throughout the Main Paper and the Supplementary Appendix or, when they did not lead to a change in the paper - and only to a clarification offered to the reviewer - in the answers to reviewers below.*

Additionally, there were a few queries by the reviewers which we were unable to address fully as they would, in the main, require access to BBVA account-level data that, at this point, we do not have available for this project. These pertain to a sub-comment in Comment 2 by Reviewer (1) (implying resolving the data at the individual level, which we

are unable to have access to for this paper); parts of Comment 1, 3, 4, and 5 of Reviewer 4 (implying access to account level deposits and savings information; price data, income data or household composition information; none of which we have access to for the purposes of this project). In each of these cases we have nevertheless engaged with the reviewers' valuable comments and provide partial answers and justifications as for why we cannot fully satisfy their requests.

Finally, regarding Reviewer #2 comments, there are two main points in the report. First, the reviewer considers the validation part of the paper as more important than the COVID-19 findings and suggests it to take center stage, by additionally expanding the scope of the paper to the years before the pandemic so as to avoid COVID-19 idiosyncrasies. Second, the reviewer considers that disclosing reasonably aggregated data (to preserve anonymity and avoid disclosing private information) is necessary for transparency in measurement.

Let us state from the outset that we are in full agreement with the Reviewer. First, we consider that the scope for bank transaction data in supplementing - and even substituting - slower moving national accounts indicators to be tantalizing. And we agree that, going forward, ambitious research efforts in validating and carefully delineating both its promise and biases will be a high-priority activity with payoffs well beyond COVID-19. Indeed, we are currently engaging with BBVA Research to pursue this ambition, though the challenges are many. Second, we firmly believe in the tenets of transparency and reproducibility and fully agree with the referee that unfettered access to data is needed to facilitate further research, both on COVID-19's socio-economic impact and further afield.

Starting with this last point, it is worth stating that originally BBVA had agreed to submit under the condition that data would not be publicly shared; rather that requests were directed to BBVA Research, which routinely collaborates with researchers worldwide, and particularly during the pandemic. Following Reviewer 2's valuable prompt - and after consultation with the RSOS Senior Publishing Editor - we have gone back to the various bank stakeholders and spent considerable time navigating legal, privacy and proprietary commercial information issues.

We are happy to report that, following efforts by the authors, BBVA now agrees to release a subset of the data under the condition that (i) this relates specifically to COVID-19 socio-economic impact (as this was the original remit of this particular project) and (ii) that both international data (due to legal jurisdiction reasons) and the most disaggregated series we use in the paper - very narrowly defined categories of card expenditure and zip-code level expenditures - are not publicly disclosed as the legal team of the bank considers that the number of bank customers in such narrow daily cells may sometimes be small enough to disclose individual level behavior. Thus, we are happy to report that following Reviewer 2's comment we now:

- Make available a replication package of codes and data which allows researchers to replicate key COVID-19 results in the paper (Figure 1, Figure 2, Figure 4 and Table 1) and to conduct their own national, subnational or expenditure-*

composition analysis in this context. Requests for all other data (aggregate historical for Spain, cross-country data, detailed category of expenditure or zip code level) should be directed to BBVA Research. Again, we reinforce that BBVA Research remains available to answer individual researchers' queries related to the remaining data used in our paper. Our data availability statement has been updated to reflect these changes.

On Reviewer 2's first point, and notwithstanding our general agreement with the referee's point (as detailed above), we have opted to keep the main part of the paper focused on COVID-19's socio-economic impact. This is because of two main reasons. Firstly, we believe that the continued prevalence of COVID-19 worldwide renders the COVID-19 analysis we conduct - on one of the most affected countries worldwide, Spain - still relevant to an interdisciplinary audience; and indeed a major reshuffling/ refocus of the paper would conflict with the recommendations of the three remaining reviewers, whose reports recommend publication in the current format. Secondly, we are at this point unable to pursue the Reviewers' suggestion to further 'expand the scope to the years before the pandemic' for our validation analysis and, in particular, this would conflict with the goal of sharing data openly and enabling researchers to replicate at least some of the analysis. This is because, as detailed in our answer to the Reviewers, for the purposes of this current submission (i) the Bank only agrees to publicly release data related to the COVID-19 period and (ii) we ourselves do not have access to detailed data (category or zip-level) pre-2019 that would enable this expanded validation.

Again, we stress that we fully agree with the reviewer that such an extended and more ambitious focus on bank transaction data outside of the COVID-19 context (and not only card transactions in this context) is likely an important contribution to the scientific community. Indeed we are currently pursuing this possibility with BBVA Research. However, the scope of that analysis, its timeline and the legal/ anonymity/ commercial issues to be resolved go well beyond what can be achieved in this current submission. Finally, our comments above notwithstanding, we note that we are unaware of another paper that conducts as much validation on bank card data as we already do here and indeed the paper is increasingly being cited on that account by researchers and policy-makers alike. Thus, we have opted to keep the bulk of our validation analysis in the paper's Supplementary Information which we see as an integral part of the paper and would remain available on the journal's website to all researchers.

Overall, thank you again for the opportunity to revise the paper for the RSOS. We hope you will be pleased with the revision.

Reviewer # 1

This is a great contribution, and I would probably publish as is. I offer three substantive comments, though.

(1) My main reservation is that I felt that this was basically two papers: one on data validation and another about COVID. Personally, I was very excited about validation of the BBVA data (I am tired of COVID), and I was disappointed that it felt like it got short shrift by relegation to Section 2 of the Appendix. The abstract promised "extensive validation exercises," and I was hoping for more. Hopefully, the lessons of covid will grow less relevant, and the BBVA (and other transaction) data will grow more prevalent!

We agree with this sentiment and are currently involved in a second project where the goal is to build national accounts from the ground up using BBVA transaction data. In this second project, we have account-level data so we can be much more careful about certain issues that this paper is unable to address, such as computing consumption series with a properly defined sampling frame and non-card spending. This is why we preferred to do basic 'sanity-check' type exercises for this paper and leave a more extensive exploration for another project. At the same time, given that COVID-19 is a historically important shock and Spain had one of the world's most dramatic initial lockdowns, we view the application as also worthy of an independent paper. Finally, we see the Supplementary Appendix as an integral part of the paper and the validation exercises we conduct there will be available on the Journal's webpage.

(2) Section 2.3 was my favorite part of the paper, and it did feel like there were a few opportunities for improvements, here.

- Rich people spend more online (Fig. 7a) -- I would love to know if they were buying more expensive stuff online, or shielding themselves from exposure for necessary transactions. (Several very old friends in Spain were doing this, but I don't know if it was as common as in the US.) Even if local supermarkets etc. were handling grocery delivery, and not big online firms, presumably you can still use whether or not the card was present at the sale (that is usually in the credit card data).

Thanks so much for this comment, as it made us think on this issue, and induced us to add a new subsection to the SI. One illustrative exercise we have done is to focus on one of the key consumption categories during the lockdown: food consumption, which in our data corresponds to categories 3-6 from Table 2 in the Supplemental Appendix. We have tabulated the total number of transactions for online food transactions during April 2019 and during April 2020 by zipcode. The figure below plots income per capita against food transactions per capita by zipcode (the number of transactions is based on an index value BBVA provided to us and is not the actual count; instead the relative values have meaning so that a zipcode with 2.0 has twice as many food transactions per capita as one with 1.0). The gradient of the regression line in 2020 is much higher than in 2019, which suggests that higher-income neighborhoods were shifting more of the food purchasing online. Thus the overall online activity of higher-income groups appears related to the purchasing of necessities.

We have added the above explanations in the SI and commented it in a footnote in the main text.

- Fig 7b: There is a lot that could be said about distances traveled for essential transactions, aka food deserts and amenities, and the authors should probably look up that literature some.

Our basic goal in section 2.3 was to show that card spending can be used as a real-time mobility proxy, and so complements the better-known mobility measures based on smartphone usage. The existence of this data permits a much deeper exploration of mobility patterns in cities, and indeed as you mention there is growing work in urban economics and other fields studying these issues. We do not view this paper as making a contribution in this space, beyond validating data at relatively high frequency. Nevertheless, we agree with your comment, and we should have cited more carefully the relevant literature. Apologies for this oversight. We now include in the reference/bibliography, two papers on the literature on this issue which may serve as a gateway to RSOS's interdisciplinary audience. One of them measures, like us, how mobility affects Covid spread (building from mobility rather than card data). The other (very relevant and related to your point) on distance travelled for different types of consumption. Rather than attempting an exhaustive list, we include these references as up-to-date papers which themselves can direct the reader to other papers in the Economics' literature. Thank you again for this suggestion.

- Distances traveled. This is where I hoped this paper was going, and it could be sharpened a bit. I was not convinced by the use of share-weighted formula (authors' line 315), in defining the distances traveled. In short, I would have summed the total activity (# of transactions) weighted PERHAPS but not necessarily by distance. If I am constructing total amount of mobility, distance is not based on shares, it is based on the number of transactions. (Caveat: of course, people string together routines, but let's not get carried away...) Just for illustration, if I am very stubborn and I insist on an espresso out, every morning, I am much more exposed than by buying a big TV to get me through the pandemic. Not only does this allow large purchases to dominate unnaturally (the location of big purchases matters more), but I think it misses the point of exposure, which is basically one per interaction. You may disagree, but then please state why.

Thank you for bringing up this valid concern. In response, we have explored the raw number of transactions as an alternative way to characterize the relationship between mobility and income. One issue is that higher-income groups on average have higher consumption, which translates into more transactions. This is one of the reasons we

conducted the original analysis in shares since this measures relative spending in different locations. Instead, when we work with raw transactions we compare total amounts in April 2019 and in April 2020 to help isolate the effect of the pandemic from that of being higher-income.

The particular exercise we conduct is to tabulate the set of “offline transactions” (i.e. those in which the card swiped a point-of-sale located in a physical shop) by postal code of cardholder residence. We further divide these into transactions that take place in the same postal code as the resident lives, and into those that take place in outside zip codes. The figures above plot these tabulations, where both transactions and income are in per-capita terms. (The number of transactions is based on an index value BBVA provided to us and is not the actual count; instead, the relative values have meaning so that a zipcode with 2.0 has twice as many food transactions per capita as one with 1.0).

For total transactions inside the home postal code, one observes almost no difference between April 2019 and April 2020. This suggests that the frequency of local shop visits did not change markedly during the pandemic relative to normal times (although the composition of spending presumably does). In contrast, there is a large difference in transactions **outside** the home postal code. The income/transaction volume gradient is much less steep during the pandemic than in 2019. In combination with the other evidence in the paper, one interpretation is that residents of higher-income postal codes were more likely to stop commuting during the lockdown. This would eliminate the outside transactions that happen during the workweek at a faster rate for higher-income people. Since outside transactions are presumably riskier in terms of disease than inside transactions, this evidence also suggests that higher-income residents were able to more effectively shield.

We have added the above explanation as a subsection to the SI, and a footnote summarizing it in the main text.

- The analysis of public and private transportation was particularly interesting, but I did not find it intuitive, mainly because the costs are different. I have no idea what 1000 EUR of car travel vs bus travel buys (line 368). You quote car as "gasoline" (not parking, and the vehicles are outside the data), so is each trip... cheaper?

Yes, this is certainly a limitation of our analysis. We have added the following footnote to acknowledge it: “We do not observe whether a unit of urban transport spending generates more or less movement through space than a unit of gasoline spending, which would also be an important input into a model of disease risk and spending.” Also, there was an error in the original text. The spending measures are all in index terms, as BBVA

did not provide the raw EUR amounts. So we now refer to generic "units" of spending rather than EUR values.

- Can you see substitution from public to private transport, perhaps more extreme in rich areas? If you have individual cardholders, you could do this at a very granular level!

Thank you for this interesting suggestion. We agree the exercise would be more convincing at an individual level; unfortunately, we do not have individual-level information available in the card dataset from which this paper builds.

- I would add another column to Tables 4 and 5 with both public transport and car spending.

Thank you for this suggestion, which we have implemented.

(3) Appendix Fig 2b: Yes, the correlations are very high, but the slope is NOT $x = y$. BBVA share is depressed at low INE share, suggesting that poorer populations do less of their spending through BBVA. Is that just because housing and fixed costs via bank transfers are a larger share? Or is it because they use cash more and/or have less access to financial services like BBVA? What does this slope imply for your estimates? Can you estimate if this difference is all "taken up" in the rents and durables? (cf app. lines 22-29).

Thank you for pointing this out. In fact, any of these explanations might be driving this result. We acknowledge this in a new footnote in the supplementary material.

(4) Typographic errors, etc.

Thank you for your detailed reading. We have corrected the mistakes you pointed out (and apologize if any remain!)

Reviewer # 2

“Comments to the Author(s)

There is a lot of rich information in this paper. Thank you. As I understand it, there are two intended contributions in this paper:

1. Show the advantages of transaction-level data over survey data for economic measurements, including national accounting.
2. Provide some empirical evidence on the effects of COVID and the lockdowns.

To summarize, I do not see the COVID part of the paper as a contribution other than to a COVID-specific issue. As for the methodological part, I can see how that can turn into a valuable contribution for the scientific community, including the supplementary material in this paper, provided the underlying data (not the micro data, but the province-level or group level, aggregated data) is released in a clean, transparent way.

Some more detailed comments:

Regarding (1), I find the context of COVID insufficient. During COVID, a lot of transactions were moved online, minimizing the common challenges of capturing a representative sample of economic activity with just one bank's information. I agree that (1) would be an important contribution, not only to Economics, but to the whole scientific community. But I would not mix it with anything regarding COVID and I'd expand the scope to the years before the pandemic, to show that these results hold more generally also when people pay in cash more often. Including the material in the Supplementary Appendix and some of the "Validation" parts in the main text.

In this case, the data release in its current form is not sufficient. Transparency in measurement is a sufficient contribution, but it has to come with transparency. I apologize in advance if this is on my end, but the data in the link (<https://www.bbvarsearch.com/en/special-section/charts>) contains only a subset of the data used in the paper. In addition, I found no way to download the data behind the graphs. Data should be provided in (csv, excel, txt..) tables to help future research. If this is indeed a problem on my end, that I did not find a way but it is there, please consider making the actual data easier to find/download.

While I understand the limitations of disclosing the underlying data, I did not see any microdata involved in the main results paper in a way that privacy would be compromised. If there is to be a paper about the measurement advantages of banking data, aggregated and anonymized data that meets usual requirements to protect privacy, fully available and not only "upon request", should be released in a clean replication package.

To point (2), the evidence presented in this paper contains some interesting and novel facts regarding behaviour of individuals around the lockdowns and pandemic, the dynamics and heterogeneity of consumption during the lockdowns, and on mobility. The different pieces of evidence, however, are relatively unstructured. I would rather focus on the methodological contribution and send the COVID part to a COVID-specific empirical or policy issue, such as <https://cepr.org/content/covid-economics-vetted-and-real-time-papers-0>.

I understand this might be too big of a change, but I recommend "major revision" in the hope that the methodological part can turn into a clean, transparent, tool for consumption measurement that would indeed be of great value to the scientific community."

Thank you very much for your insightful and in-depth comment. We fully agree with you that there are distinct contributions in this paper - the COVID findings and the more general validation of the card transaction data. Further, we fully agree that the promise of detailed bank transaction data goes well beyond this particular COVID application and that, therefore, the validation exercises may become increasingly valuable going forward, as other similar datasets are made available to researchers and COVID (hopefully!) recedes from our collective memory. Finally, we agree with you that the data we present should be made available to researchers as fully as possible.

Our agreement notwithstanding, we have opted to keep the main part of the paper devoted to the COVID shock and its consequences. This is due to the following reasons. First, we believe the findings regarding the real-time magnitude of expenditure adjustment still merit presence in peer-reviewed interdisciplinary outlets (such as the RSOS, which has continued to publish COVID-19 pieces throughout the pandemic), in particular because the findings surrounding the heterogeneous impact of lockdown policies based on types of consumption, place of residence or across the income distribution are (unfortunately) still of societal importance when thinking through and designing optimal NPIs.

Second, we not only fully agree with the reviewer that the validation of transaction data as consumption indicators is important beyond COVID, but we also believe this is important enough to merit an entirely new paper focused solely on that. This is because a systematic validation of bank transaction data will need to go well beyond the issues surrounding card transaction data that we deploy here. Without attempting to provide an exhaustive list such 'validation' paper would need to go more deeply into cash-card dynamics as the referee states; but also understanding and correcting for long-run trends in online payments; understanding the contribution and dynamics of bank (non-card) transfers and recurrent credit payments associated with durables' consumption; distinguishing between individual and household level consumption; or understanding how sample attrition affects the dynamics and volatility of the resulting time series. Further, to be clear, for the purposes of this project we were not granted access to historical granular data, that would allow us to expand further the scope of the validation exercise to the years prior to the pandemic (e.g. narrowly defined category or geographically resolved data is not available for this project for the period pre-2019), let alone the necessarily more detail account level data and individual covariates.

In short, we agree with the referee that the possibility of using bank transaction data to form national-accounts like objects (at the micro and at the macro) is tantalizing. Indeed, we are currently pursuing this research agenda together with BBVA Research and navigating the myriad of legal and commercial restrictions this implies, both inside the bank and outside. But precisely because of this, we also now know that this entails an entirely new project whose ambition - and timeline to completion - goes well beyond the validation exercises we conduct here (in the context of COVID-19 and only through the lens of card transaction data). On this point, it should also be noted that, our comments above notwithstanding, we are unaware of another paper that conducts as much validation on bank card data as we already do here and indeed the paper is increasingly being cited on that account by researchers and policy-makers alike.

Thirdly, and relatedly, repurposing the paper as a validation exercise unfortunately conflicts with your suggestion of making the maximum data available to researchers, which we also agree with wholeheartedly. In particular, following your report, we have gone back to the different data stakeholders inside BBVA to understand what data, if any, could be made public and fully available as part of a replication package. After substantial and lengthy discussions, and balancing a variety of legal, commercial and privacy concerns against the public and policy interest in accessing real-time transaction data during times of pandemics, BBVA has kindly agreed to release data concerning the Spain-wide, province-level and consumption-basket composition time series in order to allow for further research, in the context of COVID-19. On the other hand, at this time, BBVA rules out making fully available more granular data on highly disaggregated consumption categories, zip-code level data, international data or historical data outside of COVID-19 (thus precluding an extensive methodological paper outside COVID times). We stress that BBVA - and in particular BBVA Research - remains available to collaborate with researchers and answer individual requests regarding these more sensitive datasets and has a history of doing so throughout the pandemic.

Overall, in response to your comments, together with BBVA, we have reached a compromise solution. First, following your valuable prompt, BBVA now makes available (through the replication package deposited online) the national-, province- and broad category- level data, that both allows researchers to fully replicate key COVID-19 results in the paper (Figure 1, Figure 2, Figure 4 and Table 1) and to conduct their own national, subnational or compositional analysis in this context. Requests for all other data (historical for Spain, cross-country, detailed category of expenditure or zip code level) should be directed to BBVA Research. Again, we reinforce that BBVA Research remains available to answer individual researchers' queries related to the remaining data used in our paper. Our data availability statement has been updated to reflect these changes and we again thank you for your prompt on this. Second, the considerations above and the fact that the Bank is currently only available to release data enabling analysis on the impact of the COVID pandemic, led us to continue to devote the main part of the paper to the COVID shock. Nevertheless, our Supplementary Information Appendix is an integral part of the paper and remains available on the journal's website to all researchers.

Reviewer # 3

“This paper uses individual transaction data from a large Spanish bank to track consumption expenditure through the pandemic, compare the results with traditional survey-type data used by the national statistical agency, look at the effects of lockdown policies, measure mobility and estimate the relationship with external transportation expenditure, and consider differences in behaviour across income groups. The data are exceptionally interesting, the analysis is full of insights and the paper is well written. It is a first-rate piece of work, already well known in this literature, and I certainly recommend publication.”

Thank you for the kind words on our paper and for your comments and overall recommendation.

General comment.

1. There is just one general concern that I want to raise, not requiring a modification to the analysis, but rather I would suggest that the authors explain the issue more fully, so that the reader not familiar with the causal inference literature in statistics, or with the problem of endogeneity in economics, will not be misled by some of the Results. I refer primarily to the part of section 2.1 entitled ‘Effects of Lockdown and its Easing’. This section uses the language of causation (‘effects’), whereas in some other parts of this paper the authors discuss correlation and association. It is of course natural to wish to draw causal inferences about the effects of the lockdown. However, differences in the timing of the easing of restrictions across provinces are presumably not entirely exogenous—as is noted here (as selection into treatment). This nonetheless makes things more complicated than a simple natural experiment with exogenous variation. Credible attempts are made to deal with this problem, but the simple inclusion of the daily case incidence in Table 1 still leaves us in the realm of conditional expectation rather than causal inference. The difference-in-difference results are a way of attempting estimation of the treatment effect, but the conditions, assumptions and validity checks (although these are further discussed in Section 4) may deserve a more thorough explanation. It seems unlikely that other standard solutions, for example observation of conditions for a regression discontinuity design, could be put into practice here. Instead, my suggestion to the authors is that they simply describe the issues surrounding estimation of treatment effects more explicitly (ie not assume that the reader is familiar with the econometrics/ statistics of causal inference) and acknowledge any limitations which it may imply. In some cases, any bias that is created maybe in the opposite direction to the effect of interest, so that it may even be possible to argue that the observed effect is an underestimate. But in any event, I think that the reader’s attention might be drawn more directly to the difficulties involved in causal statistical inference on observational data. I emphasize that this is an expositional point only.”

Thank you for this comment. We agree with the referee that lockdown policies and the timing of province-specific lockdown easing phases should not be considered “as good as random” even when controlling for time and province fixed effects and conditional on local disease prevalence.

Following the suggestion of the referee we now devote a lengthy “word of caution” paragraph at the end of this section, explaining to a wide audience what “as good as random” causal identification would require and how the likely presence of time-varying province-specific unobservables may bias our estimates with respect to the true causal effect of lockdown and lockdown easing policies.

“Minor comments (page references are to the journal’s pdf, +2 relative to authors’ page numbering)”

“Page 7, Table 1: Please clarify the precise definitions of the dummy variables used: e.g. the matrix of province dummies has one column for each province, and within each column the variable is 1 up until the date of lockdown easing, 0 on the date of easing and thereafter. Relatedly, page 8, lines 188 and following: in columns 5 and 6 of the table, day fixed effects are added, which changes the omitted category (line 191). The signs as well as magnitudes of coefficients change in Table 1. Interpretation depends on the definitions of dummies; providing an example computation would make this much clearer.”

Thank you for this comment, our apologies if it was not clear. Following the referee’s suggestion we have now added a detailed explanation of how the province-day dummy variables are coded, additionally distinguishing the cases when there is no cross-province variation (such as the enactment of a nation-wide lockdown) vs the ones where the differential timing of lockdown easing phases introduces cross-province variation. This is included in the main text, before we discuss any of the results in Table 1. Second, also following your advice, when we turn to the specification including day and province fixed effects, we now warn the reader that this is a different specification (relying on differences in timings relative to the last common baseline) and how the interpretation and signs of the relevant coefficients change as a result of this.

Page 13 line 297: ‘during weekends *that* during working days...’ → ‘during weekends *than* during working days...’

Thank you very much for spotting this typo. We have now corrected it

“Page 16: The location of the methods section here, after the discussion section (which seems to play the role of conclusion to the paper), seems a little odd. Would it not be sensible to place methods first, wrap the main paper up with the overall discussion and conclusions, and then move to the appendices?”

We certainly agree with the referee that this would be an alternative and sensible location for methods, particularly for Economists. Nevertheless we note that this layout is a standard choice in interdisciplinary journals (like the RSOS but also other well known interdisciplinary outlets. e.g. PNAS, Nature Communications, etc) and in order to cater to the expectations of this more general audience we have decided to keep it after the discussion section.

“Page 19, discussion of the Poisson regression model. This discussion seems to pertain to the absolute number of new cases, whereas the model earlier in the text used cases per thousand. With the absolute number of new cases, there will of course be more in more highly populated areas, and yet the population of the region does not seem to be used as an explanatory variable (page 19, equation following line 524). If this is the case the omitted population variable would just project onto transportation spending, leading to a biased estimate. Is this just an incomplete description? Please clarify.”

We agree that controlling for population is important for the reasons you mention. The Poisson regression model has a panel structure, so we can (and do) include postcode fixed effects as controls. This accounts for time-invariant postcode characteristics such

as population, but also factors like the quality of the housing stock and the distance to the center of Madrid. One might argue that population is time-varying, but over the daily frequency we are considering in this exercise it can reasonably be treated as fixed.

Reviewer # 4

The Covid shock has hit economies in a similar way even if its severity differed substantially. Spain is one of the most affected countries so this kind of study that documents the significant changes in consumption patterns is extremely valuable.

There is a lot to like about this paper. The dataset the authors used is very novel, and probably the largest and most comprehensive one that exists in the current literature. By showing simple statistics and regressions, the authors make striking points. I do firmly believe that the paper would make a great contribution to the Royal Society Open Science.

I have some comments below which, I hope, would improve the paper further. I'd like to stress that these are suggestions, rather than firm revision requests. If the authors think the cost of taking these on board (due to, e.g. excessive data processing required) is significant compared to their marginal contribution, a simple discussion related to these points would suffice.

Thank you for your kind words., and for your comments and recommendations, which we have tried to implement to the best of our abilities.

1. How widely is cash used in Spain? Did this change during the pandemic? A simple statistics as to what proportion of the population has debit or credit cards would be useful. Also, the authors can look into excess deposits and check if there was a material change in the money stock during the pandemic.

We do not have any direct way to measure the share of the adult population owning debit or credit cards, but in all certainty must be very large. Debit cards are a standard issue with any bank account. In an independently obtained study from a fintech research company, The Verisk Financial Report on Spain (<https://www.veriskfinancialresearch.com>) states that:

“Payment cards are relatively well embedded among Spanish consumers, with 88% of adults holding a debit card and 55% a credit card. However, Spanish issuers have long tended not to make a clear distinction between the two, viewing cards as payment instruments or cash substitutes.”

Regarding cash usage, in our ongoing research aiming to generate National Accounts from transaction data, we have observed that cash extractions account for around 40% of payments (thus, cash being around ⅔ of direct card payments) and that this number has a markedly downward trend that increased during the pandemic, albeit it has since gone back to near-normal levels. This is inline with the above cited Verisk Financial Report which suggests that cash amounts to 67% of direct card payments.

Note that with the data we were given access to for the current RSOS paper, we did not observe all individual cash-withdrawals as many of these are done physically, at the bank branch and are not reflected in card data, but rather, in account-level data. This is why the current RSOS submission does not go into further detail; in general cash-card breakdown and both high- and low-frequency changes in this, is a complex issue and we

prefer to explore it in-depth in future research.

2. During normal times, what is the share of necessary (that households cannot cut) vs luxurious consumption (that households can cut) for high vs low-income households?

It is difficult to define what is “necessary”, and probably there is quite a bit of heterogeneity on that even controlling for income. We show in Table 3 that items like “Taxi”, “Hairdressers”, “Restaurants” or “Travel Agency” which are likely to be cataloged as “luxuries”, are correlated with the average income of the zip code. Beyond this, it is difficult to say more at this stage. In our current research, we are able to match consumption patterns to income at an individual level, and we plan to follow up in the direction that you suggest. However, this lies beyond the scope of the current RSOS paper.

One thing that we have done related to this general issue is that - motivated by your comment and the comment of another referee - we have looked at the change in online consumption of one (clear) necessity: food. We have added a subsection to the SI (and a footnote in the main text) showing that online purchases of food were more responsive to income during covid than in normal times, which is also consistent with differential change in mobility patterns across the income distribution that we document in the paper.

3. The authors have information on whether transactions take place online or offline. The natural follow-up is to look into how people substituted offline shopping with online. And if cheaper shops became more popular amongst lower-income households?

Our data do not include prices. Thus, we are unable to see whether people are substituting in the direction of cheaper brands. We agree, though, that this would be a fascinating issue to explore, but probably it would need to access supermarket level scan data or similar, which directly includes prices.

In what respects online/ offline. In the first versions of the paper, we did take a more careful look at its evolution. The following graphs are the time series of YoY growth rates separately for online and offline. They indeed show that the decline of online purchases was smaller during the lockdown (and consequently its share grew). Nevertheless, the beginning of the easing process implied a return of the share to essentially its pre-Covid level.

Moreover, in our current research using much more encompassing data, we can observe that this increase and decrease in share during the lockdown is only a blip in the more secular trend which indeed indicates a long-run growth of online and a decrease in cash transactions. We have thus opted to leave the discussion on these issues to further research.

Additionally, and as noted above, we have added a subsection to the SI looking at the

changes of online purchases of food by different income groups. During the confinement the rich increased their expenditure in online food purchases much more than the poor.

4. It might be an obvious question but debit cards are linked to customers' bank accounts, right? Isn't there a way to get their income data? Excuse the question if it's too obvious or this cannot be done due to data sharing regulations.

You are completely right. It is possible to do so, and we are currently working in this research direction. There we aim to link debit and credit card expenditure with the individual account data. The process for doing so, however, is complex and goes beyond the reach of the current RSOS paper. We find that card expenditure plus geographical information is interesting enough to deserve a paper of its own, particularly in the context of the pandemic. In any case, to be clear, we could not agree more with your comment and intend to pursue it further in future research.

5. Do the authors have information as to how big the households are? I.e. who is the main bread maker of the family or if the account is joint?

Unfortunately, it is impossible to link households in this data. Even with account data, this is far from obvious. The bank does not have explicit information on household linkages or household size. Any information has to be derived by looking at joint addresses, but that of course excludes the counting of household members without bank accounts. We are trying to find solutions to this hard problem for our next project, but it is impossible to implement anything like that in this current paper.

6. How easy for Spanish households to get a credit card? I.e. Do they have the incentive to use their debit cards more because credit cards are hard to get or are there any incentives (bonus points, cash backs) that make credit card usage more attractive? This is a minor point but credit cards are a way of pushing back household consumption, at least for a month. During the Covid crisis, some households might want to do that more than others especially if they received a big income shock.

Thanks, this is a very valid point. Debit cards are a standard-issue if you have an account, so we do not expect this to be a serious problem. Additionally, to be clear, note that the card data on this paper covers both debit and credit cards as stated in Section 4, so even if there was a substitution from debit to credit cards during the covid crisis, our data should capture the full amount of card expenditure. Unfortunately, we were not given separate identifiers/flags that allow us to distinguish whether, at purchase, a given card is credit or debit. Thus, while this breakdown would be very interesting to analyze, for the purposes of this paper, such analysis is not possible.

2 Minor comments: *Thanks a lot for this! We have corrected the typos that you pointed out.*